# Asymptotics for Sketching in Least Squares

**Edgar Dobriban**
Department of Statistics
University of Pennsylvania
Philadelphia, PA 19104
dobriban@wharton.upenn.edu

**Sifan Liu**[*]
Department of Statistics
Stanford University
Stanford, CA 94305
sfliu@stanford.edu

## Abstract

We consider a least squares regression problem where the data has been generated from a linear model, and we are interested to learn the unknown regression parameters. We consider "sketch-and-solve" methods that randomly project the data first, and do regression after. Previous works have analyzed the statistical and computational performance of such methods. However, the existing analysis is not fine-grained enough to show the fundamental differences between various methods, such as the Subsampled Randomized Hadamard Transform (SRHT) and Gaussian projections. In this paper, we make progress on this problem, working in an asymptotic framework where the number of datapoints and dimension of features goes to infinity. We find the limits of the accuracy loss (for estimation and test error) incurred by popular sketching methods. We show separation between different methods, so that SRHT is better than Gaussian projections. Our theoretical results are verified on both real and synthetic data. The analysis of SRHT relies on novel methods from random matrix theory that may be of independent interest.

## 1 Introduction

To enable learning from large datasets, randomized algorithms such as sketching or random projections are an effective approach of wide applicability (Mahoney, 2011; Woodruff, 2014; Drineas and Mahoney, 2016). In this work, we study the statistical performance of sketching algorithms in linear regression. Various versions of this fundamental problem have been studied before (see e.g., Drineas et al., 2006, 2011; Dhillon et al., 2013; Ma et al., 2015; Raskutti and Mahoney, 2016; Thanei et al., 2017, and the references therein). Specifically, in a generative model where the data are sampled from a linear regression model, Raskutti and Mahoney (2016) have recently compared the statistical performance of various sketching algorithms, such as Gaussian projections and subsampled randomized Hadamard transforms (SRHT) (introduced earlier in Sarlos (2006); Ailon and Chazelle (2006)).

However, the known results are not precise enough to enable us to distinguish between the various sketching methods. For instance, the statistical performance of Gaussian projections and SRHT is predicted to be the same (Raskutti and Mahoney, 2016), whereas the SRHT has been observed to work better in practice (Mahoney, 2011; Woodruff, 2014; Drineas and Mahoney, 2016). To address this issue, in this paper we introduce a new approach to studying sketching in least squares linear regression. As a key difference from prior work we adopt a "large-data" asymptotic limit, where the relevant dimensions and sample sizes tend to infinity, and can have arbitrary aspect ratios. By leveraging very recent results from asymptotic random matrix theory and free probability theory, we get more accurate results for the performance of sketching.

---

[*]The bulk of this work was performed while SL was a student at Tsinghua University.

Table 1: Summary of main results. We have a linear model $Y = X\beta + \varepsilon$ of size $n \times p$ and do regression after sketching on data $(SY, SX)$. We show the increase in three loss functions due to sketching: $VE$ (variance efficiency–increase in parameter estimation error), $PE$ (prediction efficiency), and $OE$ (out-of-sample prediction efficiency). The assumptions for $X$ depend on the sketching method.

| Assumption on $X$ | Arbitrary | Arbitrary | Ortho-invariant | Elliptical: $WZ\Sigma^{1/2}$ |
|---|---|---|---|---|
| Assumption on $S$ | iid entries | Haar/Hadamard | Uniform sampling | Leverage sampling |
| $VE$ | $1 + \dfrac{n-p}{r-p}$ | $\dfrac{n-p}{r-p}$ | $\dfrac{n-p}{r-p}$ | $\dfrac{\eta^{-1}_{sw^2}(1-p/n)}{\eta^{-1}_{w^2}(1-p/n)}$ |
| $PE$ | | | | $1 + \mathbb{E}[w^2(1-s)]\dfrac{\eta^{-1}_{sw^2}(1-p/n)}{p/n}$ |
| $OE$ | $\dfrac{nr-p^2}{n(r-p)}$ | $\dfrac{r(n-p)}{n(r-p)}$ | $\dfrac{r(n-p)}{n(r-p)}$ | $\dfrac{1+\mathbb{E}w^2\eta^{-1}_{sw^2}(1-\gamma)}{1+\mathbb{E}w^2\eta^{-1}_{w^2}(1-\gamma)}$ |

We study many of the most popular and important sketching methods in a unified framework, including random projection methods (Gaussian and iid projections, uniform orthogonal—Haar—projections, subsampled randomized Hadamard transforms) as well as random sampling methods (including uniform, randomized leverage-based, and greedy leverage sampling). We find clean formulas for the accuracy loss of these methods, compared to standard least squares. As an improvement over prior work, our formulas are accurate down to the constant. We verify these results in extensive simulations and on two empirical datasets.

## 1.1 Problem setup

Suppose we observe $n$ datapoints $(x_i, y_i)$, $i = 1, \ldots, n$, where $x_i$ are the $p$-dimensional features (or predictors, covariates) of the $i$-th datapoint, and $y_i$ are the continuous outcomes (or responses). We assume the usual linear model $y_i = x_i^\top \beta + \varepsilon_i$, where $\beta$ is an unknown $p$-dimensional parameter. Also $\varepsilon_i$ is the zero mean noise, with entries uncorrelated and of equal variance $\sigma^2$ across samples. In matrix form, we have $Y = X\beta + \varepsilon$, where $X$ is the $n \times p$ data matrix with $i$-th row $x_i^\top$, and $Y$ is the $n \times 1$ outcome vector with $i$-th entry $y_i$. Then the usual ordinary least squares (OLS) estimator is

$$\hat{\beta} = (X^\top X)^{-1} X^\top Y,$$

if $rank(X) = p$. This estimator is a gold standard when $n > p$, extremely popular in practice, and with many optimality properties. However, when $n, p$ are large, say on the order of millions or billions, the natural $O(np^2)$ time-complexity algorithms for computing it can be prohibitively expensive. Sketching reduces the size of the problem by multiplying $(X, Y)$ by the $r \times n$ matrix $S$ to obtain the *sketched data* $(\tilde{X}, \tilde{Y}) = (SX, SY)$. The dimensions are now $r \times p$ and $r \times 1$. Then instead of doing regression of $Y$ on $X$, we do regression of $\tilde{Y}$ on $\tilde{X}$. The solution is

$$\hat{\beta}_s = (\tilde{X}^\top \tilde{X})^{-1} \tilde{X}^\top \tilde{Y},$$

if $rank(SX) = p$. In the remainder, we assume that both $X$ and $SX$ have full column rank, which happens with probability one in the generic case if $r > p$. The computational cost decreases from $np^2$ to $rp^2$, which is significant if $r \ll n$. In parallel, the statistical error increases. There is a tradeoff between the computational cost and statistical error. The natural question is then, how much does the error increase?

**Error Criteria** To compare the statistical efficiency of the estimators $\hat{\beta}$ and $\hat{\beta}_s$, we evaluate the relative value of their mean squared error. If we use the full OLS estimator, we incur a mean squared error of $\mathbb{E}\|\hat{\beta} - \beta\|^2$. If we use the sketched OLS estimator, we incur a mean squared error of $\mathbb{E}\|\hat{\beta}_s - \beta\|^2$ instead. To see how much efficiency we lose, it is natural and customary in classical statistics to consider the relative efficiency, which is their ratio (e.g. Van der Vaart, 1998). We call this the *variance efficiency* ($VE$), because the MSE for estimation can be viewed as the sum of variances

of the OLS estimator. Hence, we define

$$VE(\hat{\beta}_s, \hat{\beta}) = \frac{\mathbb{E}\|\hat{\beta}_s - \beta\|^2}{\mathbb{E}\|\hat{\beta} - \beta\|^2}.$$

This quantity is greater than or equal to unity, so $VE \geq 1$, and *smaller is better*. An accurate sketching method would achieve an efficiency close to unity, $VE \approx 1$. Our goal will be to find VE. For completeness, we also consider the *relative prediction efficiency (PE)*, *residual efficiency (RE)*, and *out-of-sample efficiency (OE)*

$$PE = \frac{\mathbb{E}\|X\hat{\beta}_s - X\beta\|^2}{\mathbb{E}\|X\hat{\beta} - X\beta\|^2}, \quad RE = \frac{\mathbb{E}\|Y - X\hat{\beta}_s\|^2}{\mathbb{E}\|Y - X\hat{\beta}\|^2}, \quad OE = \frac{\mathbb{E}(x_t^\top \hat{\beta}_s - y_t)^2}{\mathbb{E}(x_t^\top \hat{\beta} - y_t)^2},$$

where $(x_t, y_t)$ is a test data point generated from the same model $y_t = x_t^\top \beta + \varepsilon_t$, and $x_t, \varepsilon_t$ are independent of $X, \varepsilon$, and only $x_t$ is observable. The PE quantifies the loss of accuracy in predicting the regression function $\mathbb{E}[Y|X] = X\beta$, the RE quantifies the increase in residuals, while the OE quantifies the increase in test error.

## 1.2 Our contributions

We consider a "large data" asymptotic limit, where both the dimension $p$ and the sample size $n$ tend to infinity, and their aspect ratio converges to a constant. The size $r$ of the sketched data is also proportional to the sample size. Specifically $n, p$, and $r$ tend to infinity such that the original aspect ratio converges, $p/n \to \gamma \in (0, 1)$, while the data reduction factor also converges, $r/n \to \xi \in (\gamma, 1)$. Under these asymptotics, we find the limits of the relative efficiencies under various conditions on $X$ and $S$. This asymptotic setting is different from the usual one under which sketching is studied, where $n \gg r$ (e.g., Mahoney, 2011; Woodruff, 2014; Drineas and Mahoney, 2016). However our results are accurate even in that regime. It may be possible to get convergence rates for the projections with iid entries using known results on convergence rates of Stieltjes transforms.

In practice, we do not think that $n$ or $p$ grow. Instead, for any given dataset with given $n$ and $p$, we use our results with $\gamma = p/n$ as an approximation. If $n, p$ are both relatively large (say larger than 20), then our results are already quite accurate.

It turns out that the different methods have different performance, and they are applicable to different data matrices. Our main results are summarized in Table 1. For instance, when $X$ is arbitrary and $S$ is a matrix with iid entries, the variance efficiency is $1 + (n - p)/(r - p)$, so estimation error increases by that factor due to sketching. The results are stated formally in theorems in the remainder of the paper.

**The formulas are accurate and simple**   We observe that our results are accurate, both in simulations and in two empirical data analysis examples, see Section 3. In particular, they go beyond earlier work (Raskutti and Mahoney, 2016) because they are accurate not just up to the rate, but also down to the precise constants, even in relatively small samples (see Section A.16 in the supplemental for a comparison). Moreover, they have simple expressions and do not depend on any un-estimable parameters of the data.

**Separation between sketching methods**   Our results enable us to compare the different sketching methods to a greater level of detail than previously known. For instance, in estimation error ($VE$), we have $VE_{\text{iid}} = VE_{\text{Haar}} + 1 = VE_{\text{Hadamard}} + 1$. This shows that estimation error for uniform orthogonal (Haar) random projections and the subsampled randomized Hadamard transform (SRHT) (Ailon and Chazelle, 2006) is less than for iid random projections. This shows a separation between orthogonal and iid random projections.

**Tradeoff between computation and statistical accuracy**   Each sketching method becomes more accurate as the projection dimension increases. However, this comes at an increased computational cost. We give a summary of the algorithmic complexity and statistical accuracy (variance efficiency) of each method in Section A.12, as well as a numerical comparison in Section A.17 in the supplement.

As an illustrating example, consider the dataset with $n = 10^7$ and $p = 10^5$ and we want to use SRHT before doing least squares. Our results show that if we project down to $r < n$ samples, then our test

error increases by a factor of $r(n - p)/[n(r - p)]$. Suppose now that we are willing to tolerate an increase of 1.1x in our test error. Setting $r(n - p)/[n(r - p)] = 1.1$ gives $r = 10^6$. So we can reduce the data size 10x, and only incur an increase of 1.1x in test error! This is a striking illustration of the power of sketching.

**Technical contributions**    As a specific technical contribution, our results rely on asymptotic random matrix theory (e.g., Bai and Silverstein, 2010; Couillet and Debbah, 2011; Yao et al., 2015). However, we emphasize that the "standard" results such as the Marchenko-Pastur law are *not* enough. For instance, to study the subsampled randomized Hadamard transform (SRHT), we discovered that we can use the results of (Anderson and Farrell, 2014) on *asymptotically liberating sequences*, see also (Tulino et al., 2010) for prior work. To our knowledge, this is the first time that these results are used in any statistical learning application. Given the importance of the SRHT, and the notoriously difficult nature of analyzing it, we view this as a technical innovation of broader interest.

Since there are already many different sketching methods proposed before, we do not attempt to introduce new ones here. Our goal is instead to develop a clear theory. This can lead to an increased understanding of the performance of the various methods, helping practitioners choose between them. Our theoretical framework may also help in analyzing and understanding new methods.

## 1.3    Related work

In this section we review some recent related work. Due to space limitations, we can only mention a small subset of them. For overviews of sketching and random projection methods from a numerical linear algebra perspective, see (Halko et al., 2011; Mahoney, 2011; Woodruff, 2014; Drineas and Mahoney, 2017). For a theoretical computer science perspective, see (Vempala, 2005).

(Drineas et al., 2006) show that leverage score sampling leads to better results than uniform sampling. (Drineas et al., 2012), show furthermore that leverage scores can be approximated fast using the Hadamard transform. (Drineas et al., 2011) propose the fast Hadamard transform for sketching in regression. They prove strong relative error bounds on the realized in-sample prediction error for arbitrary input data. Our results concern a different setting that assumes a generative statistical model.

One of the most related works is (Raskutti and Mahoney, 2016). They study sketching algorithms from both statistical and algorithmic perspectives. However, they focus on a different setting, where $n \gg r$, and prove bounds on $RE$ and $PE$. For instance, they discover that $RE$ can be bounded even when $r$ is not too large, proving bounds such as $RE \leq 1 + 44p/r$ for subsampling and subgaussian projections. In contrast, we show more precise results such as $|RE - r/(r - p)| = o(1)$, (without the constant 44). This holds without additional assumption for iid projections, and under the slightly stronger condition of ortho-invariance for subsampling. We show that these conditions are reasonable, because our results are accurate both in simulations and in empirical data analysis examples.

Other related works include sketching with convex constraints (Pilanci and Wainwright, 2015), column-wise sketching (Maillard and Munos, 2009; Kabán, 2014; Thanei et al., 2017), tensor sketching (Pham and Pagh, 2013; Diao et al., 2017; Malik and Becker, 2018), subspace embedding for nonlinear kernel mapping (Avron et al., 2014), partial sketching (Dhillon et al., 2013; Ahfock et al., 2017), frequent direction in streaming model (Liberty, 2013; Huang, 2018), count-min sketch (Cormode and Muthukrishnan, 2005), randomized dimension reduction in stochastic geometry (Oymak and Tropp, 2017). Sketching also has numerous applications to problems in machine learning and data science, such as clustering (Cannings and Samworth, 2017), hypothesis testing (Lopes et al., 2011), bandits (Kuzborskij et al., 2018) etc.

## 2    Theoretical results

We present our theoretical results in this section. All proofs are in the supplemental material.

## 2.1    Gaussian projection

For Gaussian random projection, the sketching matrix $S$ is generated from the Gaussian distribution. An advantage of Gaussian projections is that generating and multiplying Gaussian matrices is *embarrassingly parallel*, making it appropriate for certain distributed and cloud-computing architectures.

For the performance of Gaussian sketching, we have the following result. The first part gives exact formulas for the variance, prediction, and out-of-sample efficiencies VE, PE, and OE. The second part simplifies the OE approximation for a special class of design matrices $X$.

**Theorem 2.1** (Gaussian projection). *Suppose $S$ is an $r \times n$ Gaussian random matrix with iid standard normal entries. Let $X$ be an arbitrary $n \times p$ matrix with full column rank $p$, and suppose that $r - p > 1$. Then the efficiencies have the following form*

$$VE(\hat{\beta}_s, \hat{\beta}) = PE(\hat{\beta}_s, \hat{\beta}) = 1 + \frac{n - p}{r - p - 1},$$

$$OE(\hat{\beta}_s, \hat{\beta}) = \frac{1 + \left[ 1 + \frac{n-p}{r-p-1} \right] x_t^\top (X^\top X)^{-1} x_t}{1 + x_t^\top (X^\top X)^{-1} x_t}.$$

*Second, suppose in addition that $X$ is also random, having the form $X = Z\Sigma^{1/2}$, where $Z \in \mathbb{R}^{n \times p}$ has iid entries of zero mean, unit variance and finite fourth moment, and $\Sigma \in \mathbb{R}^{p \times p}$ is a deterministic positive definite matrix. If the test datapoint is drawn independently from the same population as $X$, i.e. $x_t = \Sigma^{1/2} z_t$, then as $n, p, r$ grow to infinity proportionally, with $p/n \to \gamma \in (0, 1)$ and $r/n \to \xi \in (\gamma, 1)$, we have the simple formula for OE*

$$\lim_{n \to \infty} OE(\hat{\beta}_s, \hat{\beta}) = \frac{\xi - \gamma^2}{\xi - \gamma} \approx \frac{nr - p^2}{n(r - p)}.$$

These results are complementary to Raskutti and Mahoney (2016), who showed that $PE \le 44(1 + n/r)$, $RE \le 1 + 44p/r$ with fixed probability under slightly different assumptions. These formulas have all the properties we claimed before: they are simple, accurate, and easy to interpret. The relative efficiencies *decrease* with $r/n$, the ratio of preserved samples after sketching. This is because a larger number of samples leads to a higher accuracy. Also, when $\xi = \lim r/n = 1$, $VE$ and $PE$ reach a minimum of 2. Thus, taking a random Gaussian projection will *degrade the performance of OLS even if we do not reduce the sample size*. This is because iid projections distort the geometry of Euclidean space due to their non-orthogonality. We will see how to overcome this using orthogonal random projections.

The proofs have three stages. The first stage, common to all sketching methods, expresses the VE and other desired quantities in terms of traces of appropriate matices. The second stage involves finding the implicit limit of those traces using random matrix theory, in terms of certain fixed-point equations from the Marchenko-Pastur law. The final stage involves finding the explicit limit. In the Gaussian case, the second and third stages simplify into explicit calculations with the Wishart distribution.

## 2.2 iid projections

For iid projections, the entries of $S$ are generated independently from the same distribution (not necessarily Gaussian). This will include *sparse projections* with iid $0, \pm 1$ entries, which can speed up computation (Achlioptas, 2001). We show that in the "large-data" limit the performance of sketching is the same as for Gaussian projections. This is an instance of *universality*.

**Theorem 2.2** (Universality for iid projection). *Suppose that $S$ has iid entries of zero mean and finite fourth moment. Suppose also that $X$ is a deterministic matrix, whose singular values are uniformly bounded away from zero and infinity. Then as $n$ goes to infinity, while $p/n \to \gamma \in (0, 1)$, $r/n \to \xi \in (\gamma, 1)$, the efficiencies have the limits*

$$\lim_{n \to \infty} VE(\hat{\beta}_s, \hat{\beta}) = \lim_{n \to \infty} PE(\hat{\beta}_s, \hat{\beta}) = 1 + \frac{1 - \gamma}{\xi - \gamma}.$$

*Suppose in addition that $X$ is also random, under the same model as in Theorem 2.1. Then the formula for OE given there still holds in this more general case.*

The proof is based on a Lindeberg exchange argument.

## 2.3 Orthogonal (Haar) random projection

We saw that a random projection with iid entries will degrade the performance of OLS *even if we do not reduce the sample size*. Matrices with iid entries are not ideal for sketching, because they distort

the geometry of Euclidean space due to their non-orthogonality. Is it possible to overcome this using orthogonal random projections? Here $S$ is a Haar random matrix uniformly distributed over the space of all $r \times n$ partial orthogonal matrices.

We need the following definition. Recall that for an $n \times p$ matrix $M$ with $n \geq p$, such that the eigenvalues of $n^{-1}M^\top M$ are $\lambda_j$, the *empirical spectral distribution (esd.)* of $M$ is the mixture $\frac{1}{p}\sum_{j=1}^{p}\delta_{\lambda_j}$, where $\delta_\lambda$ denotes a point mass distribution at $\lambda$.

**Theorem 2.3** (Haar projection). *Suppose that $S$ is an $r \times n$ Haar-distributed random matrix. Suppose also that $X$ is a deterministic matrix s.t. the esd. of $X^\top X$ converges weakly to some fixed probability distribution with compact support bounded away from the origin. Then as $n$ tends to infinity, while $p/n \to \gamma \in (0,1)$, $r/n \to \xi \in (\gamma, 1)$, the efficiencies have the limits*

$$\lim_{n\to\infty} VE(\hat{\beta}_s, \hat{\beta}) = \lim_{n\to\infty} PE(\hat{\beta}_s, \hat{\beta}) = \frac{1-\gamma}{\xi-\gamma}.$$

*Suppose in addition that the training and test data $X$ and $x_t$ are also random, under the same model as in Theorem 2.1. Then $\lim_{n\to\infty} OE(\hat{\beta}_s, \hat{\beta}) = \frac{1-\gamma}{1-\gamma/\xi}$.*

The proof uses the limiting esd of a product of Haar and fixed matrices. Orthogonal projections are *uniformly better* than iid projections in terms of statistical accuracy. For variance efficiency, $VE_{\mathrm{iid}} = VE_{\mathrm{Haar}} + 1$. However, there is still a tradeoff between statistical accuracy and computational cost, since the time complexity of generating a Haar matrix using the Gram-Schmidt procedure is $O(nr^2)$.

## 2.4 Subsampled randomized Hadamard transform

A faster way to do orthogonal projection is the subsampled randomized Hadamard transform (SRHT) (Ailon and Chazelle, 2006), also known as the Fast Johnson-Lindentsrauss transform (FJLT). This is faster as it relies on the Fast Fourier Transform, and is often viewed as a standard reference point for comparing sketching algorithms.

An $n \times n$ possibly complex-valued matrix $H$ is called a *Hadamard matrix* if $H/\sqrt{n}$ is orthogonal and the absolute values of its entries are unity, $|H_{ij}| = 1$ for $i, j = 1, \ldots, n$. A prominent example, the *Walsh-Hadamard matrix* is defined recursively by

$$H_n = \begin{pmatrix} H_{n/2} & H_{n/2} \\ H_{n/2} & -H_{n/2} \end{pmatrix},$$

with $H_1 = (1)$. This requires $n$ to be a power of 2. Another construction is the discrete Fourier transform (DFT) matrix with the $(u, v)$-th entry equal to $H_{uv} = n^{-1/2}e^{-2\pi i(u-1)(v-1)/n}$. Multiplying this matrix from the right by $X$ is equivalent to applying the discrete Fourier transform to each column of $X$, up to scaling. The time complexity for the matrix-matrix multiplication for both the transforms is $O(np \log n)$ due to the Fast Fourier Transform, faster than other random projections.

Now we consider the subsampled randomized Hadamard transform. Define the $n \times n$ subsampled randomized Hadamard matrix as $S = BHDP$, where $B \in \mathbb{R}^{n \times n}$ is a diagonal *sampling matrix* of iid Bernoulli random variables with success probability $r/n$, $H \in \mathbb{R}^{n \times n}$ is a Hadamard matrix, $D \in \mathbb{R}^{n \times n}$ is a diagonal matrix of iid random variables equal to $\pm 1$ with probability one half, and $P \in \mathbb{R}^{n \times n}$ is a uniformly distributed permutation matrix. In the definition of $S$, the Hadamard matrix $H$ is deterministic, while the other matrices $B, D$ and $P$ are random. At the last step, we discard the zero rows of $S$, so it becomes an $\tilde{r} \times n$ orthogonal matrix where $\tilde{r} \approx r$. We expect the SRHT to be similar to uniform orthogonal projections. The following theorem verifies our intuition. The proof uses free probability theory (Tulino et al., 2010; Anderson and Farrell, 2014).

**Theorem 2.4** (Subsampled randomized Hadamard projection). *Let $S$ be an $n \times n$ subsampled randomized Hadamard matrix. Suppose also that $X$ is an $n \times p$ deterministic matrix whose e.s.d. converges weakly to some fixed probability distribution with compact support bounded away from the origin. Then as $n$ tends to infinity, while $p/n \to \gamma \in (0,1)$, $r/n \to \xi \in (\gamma, 1)$, the efficiencies have the same limits as for Haar projection in Theorem 2.3.*

## 2.5 Uniform random sampling

Fast orthogonal transforms such as the Hadamard transforms are considered as a baseline for sketching methods, because they are efficient and work well quite generally. However, if the data are very

uniform, for instance if the data matrix can be assumed to be nearly rotationally invariant, then *sampling methods* can work just as well, as will be shown below.

The simplest sampling method is uniform subsampling, where we take $r$ of the $n$ rows of $X$ with equal probability, with or without replacement. Here we analyze a nearly equivalent method, where we sample each row of $X$ independently with probability $r/n$, so that the expected number of sampled rows is $r$. For large $r$ and $n$, the number of sampled rows concentrates around $r$.

Moreover, we also assume that $X$ is random, and the distribution of $X$ is *rotationally invariant*, i.e. for any $n \times n$ orthogonal matrix $U$ and any $p \times p$ orthogonal matrix $V$, the distribution of $UXV^\top$ is the same as the distribution of $X$. This holds for instance if $X$ has iid Gaussian entries. Then the following theorem states the surprising fact that uniform sampling performs just like Haar projection.

**Theorem 2.5** (Uniform sampling). *Let $S$ be an $n \times n$ diagonal uniform sampling matrix with iid Bernoulli$(r/n)$ entries. Let $X$ be an $n \times p$ rotationally invariant random matrix. Suppose that $n$ tends to infinity, while $p/n \to \gamma \in (0,1)$, and $r/n \to \xi \in (\gamma, 1)$, and the e.s.d. of $X$ converges almost surely in distribution to a compactly supported probability measure bounded away from the origin. Then the efficiencies have the same limits as for Haar matrices in Theorem 2.3.*

## 2.6 Leverage-based sampling

Uniform sampling can work poorly when the data are highly non-uniform and some datapoints are more influential than others for the regression fit. In that case, it has been proposed to sample proportionally to the leverage scores $h_{ii} = x_i^\top (X^\top X)^{-1} x_i$. These can be thought of as the "leverage of response value $Y_i$ on the corresponding value $\hat{Y}_i$". One can also do greedy leverage sampling, deterministically taking the $r$ rows with largest leverage scores (Papailiopoulos et al., 2014).

In this section, we give a unified framework to study these sampling methods. Since leverage-based sampling does not introduce enough randomness for the results to be as simple and universal as before, we need to assume some more randomness via a model for $X$. Here we consider the *elliptical model*

$$x_i = w_i \Sigma^{1/2} z_i, i = 1, \ldots, n, \tag{1}$$

where the *scale variables* $w_i$ are deterministic scalars bounded away from zero, and $\Sigma^{1/2}$ is a $p \times p$ positive definite matrix. Also, $z_i$ are iid $p \times 1$ random vectors whose entries are all iid random variables of zero mean and unit variance. This model has a long history in multivariate statistics, see (Mardia et al., 1979). If a scale variable $w_i$ is much larger than the rest, then $x_i$ will have a large leverage score. This model allows us to study the effect of unequal leverage scores. Similarly to uniform sampling, we analyze the model where each row is sampled independently with some probability.

Recall that $\eta$-transform of a distribution $F$ is defined by $\eta_F(z) = \int \frac{1}{1+zx} dF(x)$, for $z \in \mathbb{C}^+$ (e.g., Tulino and Verdú, 2004; Couillet and Debbah, 2011). In the next result, we assume that the scalars $w_i^2$, $i = 1, \ldots, n$, have a limiting distribution $F_{w^2}$ as the dimension increases. In that case, the eta-transform is the limit of the leverage scores. First we give a result for arbitrary sampling with probability $\pi_i$ depending only on $w_i$, and next specialize it to leverage sampling.

**Theorem 2.6** (Sampling for elliptical model). *Suppose $X$ is sampled from the elliptical model defined in (1). Suppose the e.s.d. of $\Sigma$ converges in distribution to some probability measure with compact support bounded away from the origin. Let $n$ tend to infinity, while $p/n \to \gamma \in (0,1)$ and $r/n \to \xi \in (\gamma, 1)$. Suppose also that the $4 + \eta$-th moment of $z_i$ is uniformly bounded, for some $\eta > 0$.*

*Consider the sketching method where we sample the $i$-th row of $X$ with probability $\pi_i$ independently, where $\pi_i$ may only depend on $w_i$, and $\pi_i, i = 1, \ldots, n$ have a limiting distribution $F_\pi$. Let $s|\pi$ be a Bernoulli random variable with success probability $\pi$, then*

$$\lim_{n\to\infty} VE(\hat{\beta}_s, \hat{\beta}) = \frac{\eta_{sw^2}^{-1}(1-\gamma)}{\eta_{w^2}^{-1}(1-\gamma)}, \quad \lim_{n\to\infty} OE(\hat{\beta}_s, \hat{\beta}) = \frac{1 + \mathbb{E}w^2 \eta_{sw^2}^{-1}(1-\gamma)}{1 + \mathbb{E}w^2 \eta_{w^2}^{-1}(1-\gamma)}$$

$$\lim_{n\to\infty} PE(\hat{\beta}_s, \hat{\beta}) = 1 + \frac{1}{\gamma} \mathbb{E}w^2(1-s)\eta_{sw^2}^{-1}(1-\gamma),$$

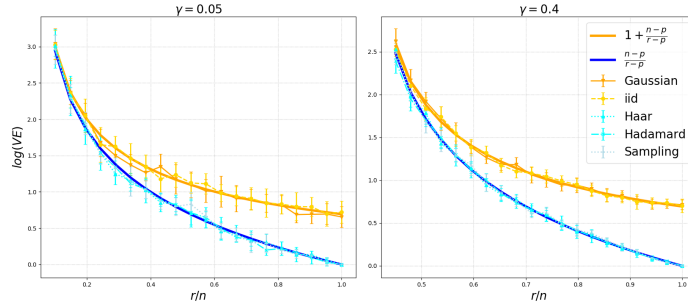

Figure 1: Verification of our theory. Solid lines show the theoretical formulas for variance efficiency, while dashed lines show the simulation results, for $\gamma = 0.05$ (left, log of VE shown), and $\gamma = 0.4$ (right). Showing SD over 10 trials of Gaussian, iid, Haar, Hadamard sketching, and sampling.

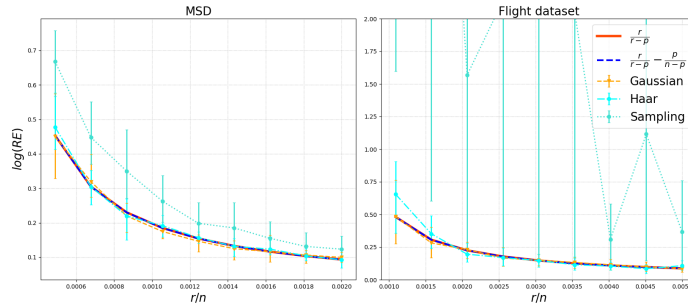

Figure 2: Empirical data analysis. Left: Million Song dataset. Right: Flight dataset.

*where $\eta_{w^2}$ and $\eta_{sw^2}$ are the $\eta$-transforms of $w^2$ (where $w$ is the distribution of scales of $x_i$) and $sw^2$ (where $s$ is defined above), respectively. Moreover, the expectation is taken with respect to the joint distribution of $s, w^2$ as defined above. In particular for leverage score sampling, $s$ is a Bernoulli variable with success probability* $\min[r/p(1 - 1/(1 + w^2 \eta_{w^2}^{-1}(1 - \gamma)), 1]$.

If $w_i$-s are all equal to unity, one can check that the results are the same as for orthogonal projection or uniform sampling on rotationally invariant $X$. This is because all leverage scores are nearly equal. We specialize this result to greedy sampling in Section A.11 in the supplement.

## 3  Simulations and data analysis

We report some simulations to verify our results. In Figure 1, we take $n = 2000$, and $p = 100$ or 800, respectively. Each row of $X$ is generated iid from $\mathcal{N}(0, I_p)$. The simulation results of VE and the error bar are the mean and one standard deviation over 10 repetitions. We also plot our theoretical results (bold lines) in the figures. The $x$-axis is on a log scale. We observe that the simulation results match the theoretical results very well. Also note that in this case, where the data is uniformly distributed, sampling methods work as well as orthogonal and Hadamard projection, while Gaussian and iid projections perform worse. Additional simulations with correlated t-distributed data and leverage sampling are in Section A.14 and A.13 in the supplement.

We test our results on the Million Song Year Prediction Dataset (MSD) (Bertin-Mahieux et al., 2011) ($n = 515344$, $p = 90$) and the New York flights dataset (Wickham, 2018) ($n = 60449$, $p = 21$). The columns are standardized to have zero mean and unit standard deviation. We compare three different sketching methods: Gaussian projection, randomized Hadamard projection, and uniform sampling. For each target dimension $r$, we show the mean, as well as 5% and 95% quantiles over 10 repetitions. The results for RE are in Figure 2, and the results for OE are in Section A.15 in the supplement. For Gaussian and Hadamard projections our theory agrees well with the experiments. However, uniform

sampling has very large variance, especially on the flight dataset. Our theory is less accurate here, because it requires the data matrix to be rotationally invariant, which may not hold.

## Discussion

A direction for future work is to study sketching in (kernel) ridge regression (perhaps possible using RMT), lasso (perhaps possible using approximate message passing). Another question is to understand the variability of sketching methods.

### Acknowledgments

The authors thank Ken Clarkson, Miles Lopes, Michael Mahoney, Mert Pilanci, Garvesh Raskutti, David Woodruff for helpful discussions. ED was partially supported by NSF BIGDATA grant IIS 1837992. SL was partially supported by a Tsinghua University Summer Research award. A version of our manuscript is available on arxiv at `https://arxiv.org/abs/1810.06089`.

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
