[Supplementary Material · supp.pdf]

# Supplement to "Asymptotics for Sketching in Least Squares Regression"

May 21, 2019

## Contents

# A   Appendix

## A.1   Mathematical background

In this section we introduce a few needed definitions from random matrix theory and free probability. See Bai and Silverstein [2010], Paul and Aue [2014], Yao et al. [2015] for references on random matrix theory and Voiculescu et al. [1992], Hiai and Petz [2006], Nica and Speicher [2006], Anderson et al. [2010] for references on free probability. The reader interested in the structure of the proofs may skip to the following sections, and refer back to this section when needed.

The data are the $n \times p$ matrix $X$ and contain $p$ features of $n$ samples. Recall that for an $n \times p$ matrix $M$ with $n \geq p$, such that the eigenvalues of $n^{-1}M^\top M$ are $\lambda_j$, the empirical spectral distribution (e.s.d.) of $M$ is the cdf of the eigenvalues. Formally, it is the mixture $\frac{1}{p}\sum_{j=1}^{p}\delta_{\lambda_j}$ where $\delta_\lambda$ denotes a point mass distribution at $\lambda$.

The aspect ratio of $X$ is $\gamma_p = p/n$. We consider limits with $p \to \infty$ and $\gamma_p \to \gamma \in (0, \infty)$. If the e.s.d. converges weakly, as $n, p, \to \infty$, to some distribution $F$, this is called the limiting spectral distribution (l.s.d.) of $X$.

The Stieltjes transform of a distribution $F$ is defined for complex valued numbers with positive imaginary part, for which $z \in \mathbb{C}^+ = \{z \in \mathbb{C} : \mathrm{Imag}(z) > 0\}$ as

$$m(z) = \int \frac{dF(x)}{x - z}.$$

This can be used to define the $S$-transform of a distribution $F$, which is a key tool for free probability. This is defined as the solution to the equation, which is unique under certain conditions (see Voiculescu et al. [1992]),

$$m_F\left(\frac{z+1}{zS(z)}\right) = -zS(z).$$

In addition to Stieltjes transform, there are other useful transforms of a distribution. The $\eta$-transform of $F$ is defined by

$$\eta_F(z) = \int \frac{1}{1+zx} dF(x) = \frac{1}{z} m_F\left(-\frac{1}{z}\right). \tag{A.1}$$

Now let us give a typical and key example of a result from asymptotic random matrix theory. Suppose the rows of $X$ are iid $p$-dimensional observations $x_i$, for $i = 1, \ldots, n$. Let $\Sigma$ be the covariance matrix of $x_i$. We consider a model of the form $X = Z\Sigma^{1/2}$, where the entries of $Z$ are iid with zero mean and unit variance, and the e.s.d. of $\Sigma$ converges weakly to a probability distribution $H$. Then the Marchenko-Pastur theorem (see Marchenko and Pastur [1967], Bai and Silverstein [2010]) states that the e.s.d. of the sample covariance matrix $n^{-1}X^\top X$ converges almost surely in distribution to a distribution $F_\gamma$, whose Stieltjes transform is the unique solution of a certain fixed point equation. A lot of information can be extracted from this equation, and we will see examples in the proofs.

Random matrix theory is related to free probability. Here we briefly introduce a few concepts in free probability that will be used in the proofs. A non-commutative probability space is a pair $(\mathcal{A}, \tau)$, where $\mathcal{A}$ is a non-commutative algebra with the unit 1 and $\tau : \mathcal{A} \to \mathbb{R}$ is a linear functional such that $\tau(1) = 1$. If $\tau(ab) = \tau(ba)$ for all $a, b \in \mathcal{A}$, then $\tau$ is called a trace. If $\tau(a^*a) \geq 0$, for all $a \in \mathcal{A}$ and the equality holds iff $a = 0$, then the trace $\tau$ is called faithful. There is also an inner product, and thus a norm, induced by $\tau$:

$$\langle a, b \rangle = \tau(a^*b), \ \|a\|^2 = \langle a, a \rangle.$$

For $a \in \mathcal{A}$ with $a = a^*$, the spectral radius $\rho(a)$ is defined by $\rho(a) = \lim_{k \to \infty} |\tau(a^{2k})|^{\frac{1}{2k}}$, whenever this limit exists. The elements in $\mathcal{A}$ are called (non-commutative) random variables, and the law (or distribution) of a random variable $a \in \mathcal{A}$ is a linear functional on the polynomial algebra $[X]$ that maps any $P(x) \in [X]$ to $\tau(P(a))$. The connection between the non-commutative probability space and classical probability theory is the spectral theorem, stating that for all $a \in \mathcal{A}$ with bounded

spectral radius, there exists a unique Borel probability measure $\mu_a$ such that for any polynomial $P(x) \in [X]$,

$$\tau(P(x)) = \int P(t)d\mu_a(t).$$

We can also define the Stieltjes transform of $a \in \mathcal{A}$ by

$$m_a(z) = \tau((a-z)^{-1}) = -\sum_{k=0}^{\infty} \frac{\tau(a^k)}{z^{k+1}},$$

which is the same as the Stieltjes transform of the probability measure $\mu_a$ associated with $a$.

Returning to random matrices, one can easily verify that

$$(\mathcal{A} = (L^{\infty-} \otimes M_n(\mathbb{R})), \tau = \frac{1}{n}\mathbb{E}\operatorname{tr})$$

is a non-commutative probability space and $\tau = \frac{1}{n}\mathbb{E}\operatorname{tr}$ is a faithful trace, where $L^{\infty-}$ denotes the collection of random variables with all moments finite. For $X \in L^{\infty-} \otimes M_n(\mathbb{R})$, the spectral radius is $\||X\|_{op}\|_{L^\infty}$, the essential supremum of the operator norm. The probability measure corresponding to the law of $X$ is the expected empirical spectral distribution

$$\mu_X = \frac{1}{n}\mathbb{E}\sum_{i=1}^{n}\delta_{\lambda_i},$$

where $\lambda_i$-s are the eigenvalues of $X$.

A collection of random variables $\{a_1, \ldots, a_k\} \subset \mathcal{A}$ are said to be freely independent (or just free) if

$$\tau[\Pi_{j=1}^{m}P_j(a_{i_j} - \tau(P_j(a_{i_j})))] = 0,$$

for any positive integer $m$, any polynomials $P_1, \ldots, P_m$ and any indices $i_1, \ldots, i_m \in [k]$ with no two adjacent $i_j$ equal Voiculescu et al. [1992], Nica and Speicher [2006]. A sequence of random variables $\{a_{1,n}, \ldots, a_{k,n}\}_{n \geq 1} \subset \mathcal{A}$ is said to be asymptotically free if

$$\tau[\Pi_{j=1}^{m}P_j(a_{i_j,n} - \tau(P_j(a_{i_j,n})))] \to 0,$$

for any positive integer $m$, any polynomials $P_1, \ldots, P_m$ and any indices $i_1, \ldots, i_m \in [k]$ with no two adjacent $i_j$ equal. If $a, b \in \mathcal{A}$ are free, then the law of their product is called their freely multiplicative convolution, and is denoted $a \boxtimes b$.

A fundamental result is that the $S$-transform of $a \boxtimes b$ equals the products of $S_a(z)$ and $S_b(z)$ Voiculescu et al. [1992], Nica and Speicher [2006]. In addition, random matrices with sufficiently independent entries and "near-uniformly" distributed eigenvectors tend to be asymptotically free in the high-dimensional limit. This is a powerful tool to find the l.s.d. of a product of random matrices.

## A.2   Finite-sample results for fixed matrices

We start with finite-sample results that are true for any fixed sketching matrix $S$. These results will be fundamental in all remaining work. Later, to simplify these results, we will make probabilistic assumptions. First we find a more explicit form of the relative efficiencies.

**Proposition A.1** (Finite $n$ results). *Taking expectations only over the noise $\varepsilon$ and $\varepsilon_t$, fixing $X$ and $S$, the efficiencies have the following forms:*

$$VE(\hat{\beta}_s, \hat{\beta})|X, S = \frac{\mathrm{tr}[Q_1]}{\mathrm{tr}[(X^\top X)^{-1}]}, \quad PE(\hat{\beta}_s, \hat{\beta})|X, S = \frac{\mathrm{tr}[Q_2]}{p},$$

$$OE(\hat{\beta}_s, \hat{\beta})|X, S = \frac{1 + x_t^\top Q_1 x_t}{1 + x_t^\top (X^\top X)^{-1} x_t},$$

*where $Q_0 = (X^\top S^\top S X)^{-1} X^\top S^\top S$, while $Q_1 = Q_0 Q_0^\top$, and $Q_2 = X Q_1 X^\top$.*

*Proof.* The OLS before and after sketching give the estimators $\hat{\beta}$ and $\hat{\beta}_s$

$$\hat{\beta}_{full} = (X^\top X)^{-1} X^\top Y = \beta + (X^\top X)^{-1} X^\top \varepsilon,$$
$$\hat{\beta}_{sub} = (\tilde{X}^\top \tilde{X})^{-1} \tilde{X}^\top \tilde{Y} = \beta + (\tilde{X}^\top \tilde{X})^{-1} \tilde{X}^\top \varepsilon = \beta + Q_0 \varepsilon.$$

We define the "hat" matrices

$$H = X(X^\top X)^{-1} X^\top,$$
$$\tilde{H} = X(\tilde{X}^\top \tilde{X})^{-1} \tilde{X}^\top S = X(X^\top S^\top S X)^{-1} X^\top S^\top S = X Q_0.$$

These are both projection matrices, i.e., they satisfy the relations $H^2 = H, \tilde{H}^2 = \tilde{H}$. By our assumptions, we have that $\mathbb{E}_\varepsilon [\varepsilon] = 0_n$, $\mathbb{E}_\varepsilon [\varepsilon \varepsilon^\top] = \sigma^2 I_n$, $\mathrm{tr}[H] = \mathrm{tr}[\tilde{H}] = p$. Therefore, we can calculate as follows.

1. Variance efficiency:

$$\mathbb{E}_\varepsilon \left[\|\hat{\beta} - \beta\|^2\right] = \mathbb{E}_\varepsilon \left[\|(X^\top X)^{-1} X^\top \varepsilon\|^2\right] = \sigma^2 \mathrm{tr}[(X^\top X)^{-1}]$$
$$\mathbb{E}_\varepsilon \left[\|\hat{\beta}_s - \beta\|^2\right] = \mathbb{E}_\varepsilon \left[\|Q_0 \varepsilon\|^2\right] = \sigma^2 \mathrm{tr}(Q_0 Q_0^\top).$$

   This proves the formula for $VE$.

2. Prediction efficiency:

$$\mathbb{E}_\varepsilon \left[\|X\beta - X\hat{\beta}\|^2\right] = \mathbb{E}_\varepsilon \left[\|H\varepsilon\|^2\right] = \sigma^2 \mathrm{tr}[H] = p\sigma^2,$$
$$\mathbb{E}_\varepsilon \left[\|X\beta - X\hat{\beta}_s\|^2\right] = \mathbb{E}_\varepsilon \left[\|\tilde{H}\varepsilon\|^2\right] = \sigma^2 \mathrm{tr}[\tilde{H}^\top \tilde{H}] = \sigma^2 \mathrm{tr}(Q_2).$$

   This finishes the calculation for $PE$.

3. Out-of-sample efficiency:

$$\mathbb{E}_{\varepsilon, \varepsilon_t} \left[(y_t - x_t^\top \hat{\beta})^2\right] = \mathbb{E}_{\varepsilon, \varepsilon_t} \left[(\varepsilon_t - x_t^\top (X^\top X)^{-1} X^\top \varepsilon)^2\right]$$
$$= \mathbb{E}_{\varepsilon, \varepsilon_t} \left[\varepsilon_t^2 + \varepsilon^\top X(X^\top X)^{-1} x_t x_t^\top (X^\top X)^{-1} X^\top \varepsilon\right]$$
$$= \sigma^2 (1 + x_t^\top (X^\top X)^{-1} x_t),$$
$$\mathbb{E}_{\varepsilon, \varepsilon_t} \left[(y_t - x_t^\top \hat{\beta}_s)^2\right] = \mathbb{E}_{\varepsilon, \varepsilon_t} \left[(\varepsilon_t - x_t^\top Q_0 \varepsilon)^2\right] = \sigma^2 (1 + x_t^\top Q_0 Q_0^\top x_t).$$

This finishes the proof. □

The expressions simplify considerably for orthogonal matrices $S$. Suppose that $S$ is an $r \times n$ matrix such that $SS^\top = I_r$, then we have the following result:

**Proposition A.2** (Finite $n$ results for orthogonal $S$). *When $S$ is an orthogonal matrix, the above formulas simplify to*

$$VE = \frac{\mathrm{tr}[(X^\top S^\top S X)^{-1}]}{\mathrm{tr}[(X^\top X)^{-1}]}, \quad PE = \frac{\mathrm{tr}[(X^\top S^\top S X)^{-1} X^\top X]}{p},$$

$$OE = \frac{1 + x_t^\top (X^\top S^\top S X)^{-1} x_t}{1 + x_t^\top (X^\top X)^{-1} x_t}.$$

*Proof.* Since $S$ satisfies $SS^\top = I_r$, we have $(S^\top S)^2 = S^\top S$. Thus, $Q_1 = Q_0 Q_0^\top = (X^\top S^\top S X)^{-1}$. With this, the results follow directly from Proposition A.1. □

Actually these formulas hold for any $S$ s.t. $X^\top S^\top S X$ is nonsingular and $S^\top S$ is idempotent.

## A.3 Proof of Theorem 2.1

The proof below utilizes the orthogonal invariance of Gaussian matrices and properties of Wishart matrices. For any $X \in \mathbb{R}^{n \times p}$ with $n \geq p$ and with full column rank, we have the singular value decomposition (SVD) $X = U \Lambda V^\top$, where $U \in \mathbb{R}^{n \times p}, V \in \mathbb{R}^{p \times p}$ are both orthogonal matrices, while $\Lambda \in \mathbb{R}^{p \times p}$ is a diagonal matrix, whose diagonal entries are the singular values of $X$. Therefore

$$VE(\hat{\beta}_s, \hat{\beta}) = \frac{\mathbb{E}\left[\mathrm{tr}((X^\top S^\top S X)^{-2} X^\top (S^\top S)^2 X)\right]}{\mathbb{E}\left[\mathrm{tr}[(X^\top X)^{-1}]\right]}$$

$$= \frac{\mathbb{E}\left[\mathrm{tr}(\Lambda^{-2}(U^\top S^\top S U)^{-1} U^\top (S^\top S)^2 U (U^\top S^\top S U)^{-1})\right]}{\mathbb{E}\left[\mathrm{tr}(\Lambda^{-2})\right]},$$

$$PE(\hat{\beta}_s, \hat{\beta}) = \frac{\mathbb{E}\left[\mathrm{tr}((X^\top S^\top S X)^{-1} X^\top X (X^\top S^\top S X)^{-1} X^\top (S^\top S)^2 X)\right]}{p}$$

$$= \frac{\mathbb{E}\left[\mathrm{tr}((U^\top S^\top S U)^{-2} U^\top (S^\top S)^2 U)\right]}{p},$$

$$OE(\hat{\beta}_s, \hat{\beta}) = \frac{1 + \mathbb{E}\left[x_t^\top (X^\top S^\top S X)^{-1} X^\top (S^\top S)^2 X (X^\top S^\top S X)^{-1} x_t\right]}{1 + \mathbb{E}\left[x_t^\top (X^\top X)^{-1} x_t\right]}$$

$$= \frac{1 + \mathbb{E}\left[x_t^\top V \Lambda^{-1} (U^\top S^\top S U)^{-1} U^\top (S^\top S)^2 U (U^\top S^\top S U)^{-1} \Lambda^{-1} V^\top x_t\right]}{1 + \mathbb{E}\left[x_t^\top V \Lambda^{-2} V^\top x_t\right]}.$$

We can see that the first two relative efficiencies do not depend on the right singular vectors of $X$.

We denote by $U^\perp \in \mathbb{R}^{n \times (n-p)}$ a complementary orthogonal matrix of $U$, such that $UU^\top + U^\perp U^{\perp\top} = I_n$. Let $S_1 = SU$, $S_2 = SU^\perp$, of sizes $r \times p$, and $r \times (n-p)$, respectively. Then $S_1$ and $S_2$ both have iid $\mathcal{N}(0,1)$ entries and they are independent from each other, because of the orthogonal invariance of a Gaussian random matrix. Also note that

$$SS^\top = S(UU^\top + U^\perp U^{\perp\top})S^\top = S_1 S_1^\top + S_2 S_2^\top,$$

and

$$S_1^\top S_1 \sim \mathcal{W}_p(I_p, r), \quad S_2 S_2^\top \sim \mathcal{W}_r(I_r, n-p),$$

where $\mathcal{W}_p(\Sigma, r)$ is the Wishart distribution with $r$ degrees of freedom and scale matrix $\Sigma$. Then by the properties of Wishart distribution [e.g., Anderson, 2003], when $r - p > 1$, we have

$$\mathbb{E}\left[(S_1^\top S_1)^{-1}\right] = \frac{I_p}{r-p-1}, \ \mathbb{E}\left[S_2 S_2^\top\right] = (n-p)I_r.$$

Hence the numerator of $VE$ equals

$$\begin{aligned}
&\mathbb{E}\left[\operatorname{tr}\left(\Lambda^{-2}(U^\top S^\top SU)^{-1}U^\top(S^\top S)^2 U(U^\top S^\top SU)^{-1}\right)\right] \\
&= \mathbb{E}\left[\operatorname{tr}\left(\Lambda^{-2}(S_1^\top S_1)^{-1}S_1(S_1 S_1^\top + S_2 S_2^\top)S_1^\top(S_1^\top S_1)^{-1}\right)\right] \\
&= \operatorname{tr}\left(\Lambda^{-2}(I_p + \mathbb{E}\left[(S_1^\top S_1)^{-1}S_1^\top S_2 S_2^\top S_1(S_1^\top S_1)^{-1}\right])\right) \\
&= \operatorname{tr}\left(\Lambda^{-2}(I_p + \mathbb{E}\left[(S_1^\top S_1)^{-1}S_1^\top(n-p)I_p S_1(S_1^\top S_1)^{-1}\right])\right) \\
&= \operatorname{tr}\left(\Lambda^{-2}(I_p + (n-p)\mathbb{E}\left[(S_1^\top S_1)^{-1}\right])\right) \\
&= \operatorname{tr}\left(\Lambda^{-2}(1 + \frac{n-p}{r-p-1})\right),
\end{aligned}$$

and the denominator $\operatorname{tr}[(X^\top X)^{-1}] = \operatorname{tr}[(V\Lambda^2 V^\top)^{-1}] = \operatorname{tr}(\Lambda^{-2})$, so we have $VE(\hat{\beta}_s, \hat{\beta}) = 1 + \frac{n-p}{r-p-1}$. This finishes the calculation for VE. See Section A.5 for the remaining details of this theorem.

## A.4 Proof of Theorem 2.2

The proof idea is to use a Lindeberg swapping argument to show that the results from Gaussian matrices extend to iid matrices provided that the first two moments match.

Since the error criteria are invariant under the scaling of $S$, we can assume without loss of generality that the entries of $S$ are $n^{-1/2}s_{ij}$, where $s_{ij}$ are iid random variables of zero mean, unit variance, and finite fourth moment. We also let $T = n^{-1/2}t_{ij}$, $t_{ij}$ being iid standard Gaussian random variables, for all $i \in [r]$, $j \in [n]$.

Let $s$ (respectively, $t$) be the $rn$-dimensional vector whose entries are $s_{ij}$ (respectively, $t_{ij}$) aligned by columns. Then there is a bijection from $s$ to $S$, and from $t$ to $T$. We already know that the desired results for $VE$ and $PE$ hold if $S = T$, and they only depend on $\mathbb{E}\left[\operatorname{tr}(Q_1)\right]$ and $\mathbb{E}\left[\operatorname{tr}(Q_2)\right]$.

For $OE$, under the extra assumptions that $X = Z\Sigma^{1/2}$, we already proved in Theorem 2.1 that

$$\mathbb{E}\left[x_t^\top(\frac{1}{p}X^\top X)^{-1}x_t\right] - \operatorname{tr}[(\frac{1}{p}Z^\top Z)^{-1}] \xrightarrow{a.s.} 0,$$

$$\mathbb{E}\left[x_t^\top Q_1 x_t\right] - \operatorname{tr}(Q_1) \xrightarrow{a.s.} 0,$$

so the results for $OE$ will only depend on $\mathbb{E}\left[\operatorname{tr}[Q_1]\right]$ as well. Thus we only need to show that $\mathbb{E}\left[\operatorname{tr}[Q_1(S, X)]\right]$ has the same limit as $\mathbb{E}\left[\operatorname{tr}[Q_1(T, X)]\right]$, and $\mathbb{E}\left[\operatorname{tr}[Q_2(S, X)]\right]$ has the same limit as $\mathbb{E}\left[\operatorname{tr}[Q_2(T, X)]\right]$, as $n$ goes to infinity.

Since $SX$ has a nonzero chance of being singular, it is necessary first to show the universality for a regularized trace. See Section A.6.1 for the proof of Lemma A.3 below. In the rest of the proof, we let $N = rn$.

**Lemma A.3** (Universality for regularized trace functionals). *Let $z_n = \frac{i}{n} \in \mathbb{C}$, where $i$ is the imaginary unit. Define the functions $f_N, g_N : \mathbb{R}^N \to \mathbb{R}$ as*

$$f_N(s) = \frac{1}{p} \operatorname{tr}[(X^\top S^\top S X - z_n I_p)^{-2} X^\top (S^\top S)^2 X], \tag{A.2}$$

$$g_N(s) = \frac{1}{p} \operatorname{tr}[(X^\top S^\top S X - z_n I_p)^{-1} X^\top X (X^\top S^\top S X - z_n I_p)^{-1} X^\top (S^\top S)^2 X], \tag{A.3}$$

*Then $\lim_{n \to \infty} |\mathbb{E}[f_N(s)] - \mathbb{E}[f_N(t)]| = 0$, $\lim_{n \to \infty} |\mathbb{E}[g_N(s)] - \mathbb{E}[g_N(t)]| = 0$.*

Next we show that the regularized trace functionals have the same limit as the ones we want. See Section A.6.2 for the proof.

**Lemma A.4** (Convergence of trace functionals). *Define the functions $f_\infty, g_\infty : \mathbb{R}^N \to \mathbb{R}$*

$$f_\infty(s) = \frac{1}{p} \operatorname{tr}[(X^\top S^\top S X)^{-2} X^\top (S^\top S)^2 X] = \frac{1}{p} \operatorname{tr}[Q_1(S, X)], \tag{A.4}$$

$$g_\infty(s) = \frac{1}{p} \operatorname{tr}[(X^\top S^\top S X)^{-1} X^\top X (X^\top S^\top S X)^{-1} X^\top (S^\top S)^2 X] = \frac{1}{p} \operatorname{tr}[Q_2(S, X)]. \tag{A.5}$$

*Then*

$$\lim_{n \to \infty} |\mathbb{E}[f_N(s)] - \mathbb{E}[f_\infty(s)]| = \lim_{n \to \infty} |\mathbb{E}[f_N(t)] - \mathbb{E}[f_\infty(t)]| = 0,$$
$$\lim_{n \to \infty} |\mathbb{E}[g_N(s)] - \mathbb{E}[g_\infty(s)]| = \lim_{n \to \infty} |\mathbb{E}[g_N(t)] - \mathbb{E}[g_\infty(t)]| = 0.$$

According to lemma A.3 and A.4, we know that

$$\lim_{n \to \infty} \frac{1}{p} \mathbb{E}[\operatorname{tr}[Q_1(S, X)]] = \lim_{n \to \infty} \frac{1}{p} \mathbb{E}[\operatorname{tr}[Q_1(T, X)]],$$
$$\lim_{n \to \infty} \frac{1}{p} \mathbb{E}[\operatorname{tr}[Q_2(S, X)]] = \lim_{n \to \infty} \frac{1}{p} \mathbb{E}[\operatorname{tr}[Q_2(T, X)]],$$

which concludes the proof of Theorem 2.2.

## A.5   Proof of Theorem 2.1

For the numerator of $OE$, note that

$$\mathbb{E}\left[x_t^\top V \Lambda^{-1} (U^\top S^\top S U)^{-1} U^\top (S^\top S)^2 U (U^\top S^\top S U)^{-1} \Lambda^{-1} V^\top x_t\right]$$
$$= \operatorname{tr}[\mathbb{E}\left[(S_1^\top S_1)^{-1} S_1^\top (S_1 S_1^\top + S_2 S_2^\top) S_1 (S_1^\top S_1)^{-1}\right] \Lambda^{-1} V^\top x_t x_t^\top V \Lambda^{-1}]$$
$$= \operatorname{tr}[(I_p + \mathbb{E}\left[(S_1^\top S_1)^{-1} S_1^\top (n - p) I_r S_1 (S_1^\top S_1)^{-1}\right]) \Lambda^{-1} V^\top x_t x_t^\top V \Lambda^{-1}]$$
$$= \operatorname{tr}[(I_p + \frac{n - p}{r - p - 1} I_p) \Lambda^{-1} V^\top x_t x_t^\top V \Lambda^{-1}]$$
$$= (1 + \frac{n - p}{r - p - 1}) x_t^\top V \Lambda^{-2} V^\top x_t.$$

Therefore

$$OE(\hat{\beta}_s, \hat{\beta}) = \frac{1 + (1 + \frac{n-p}{r-p-1})x_t^\top (X^\top X)^{-1} x_t}{1 + x_t^\top (X^\top X)^{-1} x_t}.$$

Additionally, if $x_t = \Sigma^{1/2} z_t$ and $X = Z\Sigma^{1/2}$, we have $x_t^\top (X^\top X)^{-1} x_t = z_t^\top (Z^\top Z)^{-1} z_t$. Since $z_t$ has iid entries of zero mean and unit variance, we have

$$\mathbb{E}\left[z_t^\top (Z^\top Z)^{-1} z_t\right] = \text{tr}[\mathbb{E}\left[(Z^\top Z)^{-1}\right] \mathbb{E}\left[z_t z_t^\top\right]] = \text{tr}[\mathbb{E}\left[(Z^\top Z)^{-1}\right]]$$

Note that the e.s.d. of $\frac{1}{n}Z^\top Z$ converges almost surely to the standard $Mar\check{c}enko - Pastur$ law [Marchenko and Pastur, 1967, Bai and Silverstein, 2010] whose Stieltjes transform $m(z)$ satisfies the equation

$$m(z) = \frac{1}{1 - \gamma - z - z\gamma m(z)}$$

for $z \notin [(1 - \sqrt{\gamma})^2, (1 + \sqrt{\gamma})^2]$. Letting $z = 0$, we have $m(0) = 1/(1-\gamma)$, thus

$$\text{tr}[(\frac{1}{n}ZZ^\top)^{-1}] \xrightarrow{a.s.} \frac{1}{1-\gamma}, \quad \text{tr}[(\frac{1}{p}ZZ^\top)^{-1}] \xrightarrow{a.s.} \frac{\gamma}{1-\gamma}.$$

Therefore $\mathbb{E}\left[x_t^\top (X^\top X)^{-1} x_t\right] \xrightarrow{a.s.} \frac{\gamma}{1-\gamma}$ and almost surely

$$OE(\hat{\beta}_s, \hat{\beta}) \to \frac{1 + (1 + \frac{1-\gamma}{\xi-\gamma})\frac{\gamma}{1-\gamma}}{1 + \frac{\gamma}{1-\gamma}} = \frac{\xi - \gamma^2}{\xi - \gamma}, \text{ as } n \to \infty.$$

Similarly for the numerator of $PE$, we have

$$\begin{aligned}
\mathbb{E}\left[\text{tr}((U^\top S^\top S U)^{-2} U^\top (S^\top S)^2 U)\right] &= \mathbb{E}\left[\text{tr}((S_1^\top S_1)^{-2} S_1^\top (S_1 S_1^\top + S_2 S_2^\top) S_1)\right] \\
&= \mathbb{E}\left[\text{tr}(I_p + (S_1^\top S_1)^{-2} S_1^\top S_2 S_2^\top S_1)\right] \\
&= p + \text{tr}(\mathbb{E}\left[(S_1^\top S_1)^{-2} S_1^\top (n-p) I_r S_1\right]) \\
&= p + (n-p)\text{tr}(\mathbb{E}\left[(S_1^\top S_1)^{-1}\right]) \\
&= p + \frac{(n-p)p}{r-p-1},
\end{aligned}$$

therefore

$$PE(\hat{\beta}_s, \hat{\beta}) = \frac{p + \frac{(n-p)p}{r-p-1}}{p} = 1 + \frac{n-p}{r-p-1}.$$

This finishes the proof.

## A.6 Proof of Theorem 2.2

### A.6.1 Proof of Lemma A.3

The proof of this lemma relies on the Lindeberg Principle, similar to the Generalized Lindeberg Principle, Theorem 1.1 of Chatterjee [2006]. The first claim shows universality assuming bounded third derivatives.

**Lemma A.5** (Universality theorem). *Suppose $s$ and $t$ are two independent random vectors in $\mathbb{R}^N$ with independent entries, satisfying $\mathbb{E}\left[s_i\right] = \mathbb{E}\left[t_i\right]$ and $\mathbb{E}\left[s_i^2\right] = \mathbb{E}\left[t_i^2\right]$ for all $1 \leq i \leq N$, and $\mathbb{E}\left[|s_i|^3 + |t_i|^3\right] \leq M < \infty$. Suppose $f_N \in C^3(\mathbb{R}^N, \mathbb{R})$ and $|\frac{\partial^3 f_N}{\partial s_i^3}|$ is bounded above by $L_N$ for all $1 \leq i \leq N$ and almost surely as $N$ goes to infinity, then*

$$|\mathbb{E}\left[f_N(s) - f_N(t)\right]| = O(L_N N), \ \ as \ N \to \infty.$$

The lemma below shows that the third derivatives are actually bounded for our functions of interest, and that the $L_N$ are of order $N^{-3/2}$.

Since we know the singular values of $X$ are uniformly bounded away from zero and infinity, there exists a constant $c > 0$, such that

$$\frac{1}{c} \leq \sigma_{\min}(X) \leq \sigma_{\max}(X) \leq c.$$

**Lemma A.6** (Bounding the third derivatives). *Let $f_N(s)$ and $g_N(s)$ be defined in (A.2) and (A.3), where the entries of $s$ are independent, of zero mean, unit variance and finite fourth moment. Then there exists some constant $\phi = \phi(c, \xi, \gamma) > 0$, such that for any partial derivative $\partial_\alpha = \frac{\partial}{\partial_{ij}}$, $\forall i \in [r], j \in [n]$,*

$$|\partial_\alpha^3 f_N| \leq \phi N^{-5/4}, \quad |\partial_\alpha^3 g_N| \leq \phi N^{-5/4}$$

*hold almost surely as $n$ goes to infinity.*

The above two lemmas conclude the proof of Lemma A.3. Next we prove them in turn.

*Proof.* (Proof of Lemma A.5) The main idea of this proof is borrowed from the proof of Theorem 1.1 of Chatterjee [2006]. For each fixed $N$, We write

$$s = (s_1, \ldots, s_N), \quad t = (t_1, \ldots, t_N).$$

For each $i = 0, 1, \ldots, N$, define

$$z_i = (s_1, \ldots, s_{i-1}, s_i, t_{i+1}, \ldots, t_N),$$
$$z_i^0 = (s_1, \ldots, s_{i-1}, 0, t_{i+1}, \ldots, t_N).$$

Note that $z_0 = t, z_N = s$. By a Taylor expansion, we have almost surely that

$$|f_N(z_i) - f_N(z_i^0) - \partial_i f_N(z_i^0)s_i - \frac{1}{2}\partial_i^2 f_N(z_i^0)s_i^2| \leq \frac{1}{6}L_N|s_i|^3,$$
$$|f_N(z_{i-1}) - f_N(z_i^0) - \partial_i f_N(z_i^0)t_i - \frac{1}{2}\partial_i^2 f_N(z_i^0)t_i^2| \leq \frac{1}{6}L_N|t_i|^3.$$

Thus

$$|f_N(z_i) - f_N(z_{i-1}) - \partial_i f_N(z_i^0)(s_i - t_i) - \frac{1}{2}\partial_i^2 f_N(z_i^0)(s_i^2 - t_i^2)| \leq \frac{1}{6}(|s_i|^3 + |t_i|^3)L_N.$$

Since

$$f_N(s) - f_N(t) = \sum_{i=1}^{N} f_N(z_i) - f_N(z_{i-1}),$$

we have

$$|f_N(s) - f_N(t) - \sum_{i=1}^{N} \partial_i f_N(z_i^0)(s_i - t_i) - \sum_{i=1}^{N} \frac{1}{2}\partial_i^2 f_N(z_i^0)(s_i^2 - t_i^2)| \leq \sum_{i=1}^{N} \frac{1}{6}(|s_i|^3 + |t_i|^3)L_N$$

almost surely as $N$ goes to infinity. By the bounded convergence theorem, and because the first two moments of $s, t$ match, we have

$$|\mathbb{E}\left[f_N(s) - f_N(t)\right]| \leq \frac{1}{6}\mathbb{E}\left[(|s_i|^3 + |t_i|^3)\right]L_N N,$$

thus

$$|\mathbb{E}\left[f_N(s) - f_N(t)\right]| \leq O(L_N N).$$

This proves Lemma A.5. □

*Proof.* (Proof of Lemma A.6) We will show that the third derivative of $f_N$ and $g_N$ are both bounded in magnitude by $N^{-5/4}$, or equivalently, $n^{-5/2}$. For any $\alpha = (i,j) \in [r] \otimes [n]$, denote $\partial_\alpha = \frac{\partial}{\partial_{ij}}$. Define

$$G_n(S) = (X^\top S^\top S X - z_n I_p)^{-2} X^\top (S^\top S)^2 X,$$

then we have $f_N(s) = \frac{1}{p}\operatorname{tr}(G_n(S))$ and

$$(X^\top S^\top S X - z_n I_p)^2 G_n(S) = X^\top (S^\top S)^2 X. \tag{A.6}$$

Take derivative w.r.t. $\alpha$ on both sides and we get

$$\partial_\alpha[(X^\top S^\top S X - z_n I_p)^2] \cdot G_n(S) + (X^\top S^\top S X - z_n I_p)^2 \cdot \partial_\alpha G_n(S) = \partial_\alpha[X^\top (S^\top S)^2 X]. \tag{A.7}$$

We have

$$\begin{aligned}
\partial_\alpha[(X^\top S^\top S X - z_n I_p)^2] &= \partial_\alpha[(X^\top S^\top S X)^2] - 2z_n\partial_\alpha(X^\top S^\top S X) \\
&= \partial_\alpha(X^\top S^\top S X) \cdot (X^\top S^\top S X) + (X^\top S^\top S X) \cdot \partial_\alpha(X^\top S^\top S X) \\
&\quad - 2z_n\partial_\alpha(X^\top S^\top S X),
\end{aligned}$$

and

$$\partial_\alpha(X^\top S^\top S X) = X^\top[\partial_\alpha(S^\top) \cdot S + S^\top \cdot \partial_\alpha S]X = X^\top(n^{-1/2}E_{ji}S + S^\top n^{-1/2}E_{ij})X,$$

where $E_{ij} \in \mathbb{R}^{r \times n}$ whose $(i,j)$-th entry is 1 and the rest are all zeros, and $E_{ji} = E_{ij}^\top$. Therefore

$$\begin{aligned}
\partial_\alpha[(X^\top S^\top S X - z_n I_p)^2] &= [X^\top(E_{ji}S + S^\top E_{ij})XX^\top S^\top S X + X^\top S^\top S X X^\top(E_{ji}S + S^\top E_{ij})X \\
&\quad - 2z_n X^\top(E_{ji}S + S^\top E_{ij})X]n^{-1/2}.
\end{aligned} \tag{A.8}$$

Similarly,

$$
\begin{aligned}
\partial_\alpha[X^\top(S^\top S)^2 X] &= X^\top[\partial_\alpha(S^\top S)\cdot(S^\top S) + (S^\top S)\cdot\partial_\alpha(S^\top S)]X \\
&= \left\{X^\top[(E_{ji}S + S^\top E_{ij})(S^\top S) + (S^\top S)(E_{ji}S + S^\top E_{ij})]X\right\}n^{-1/2}. \quad\text{(A.9)}
\end{aligned}
$$

Denoting $P(S) = X^\top S^\top S X$ and $Q(S) = E_{ji}S + S^\top E_{ij}$, substituting (A.8),(A.9) into (A.7), we get

$$
\begin{aligned}
\partial_\alpha G(S) = (P(S) - z_n I_p)^{-2}&\{X^\top[Q(S)S^\top S + S^\top S Q(S)]X \\
&- [X^\top Q(S)X P(S) + P(S)X^\top Q(S)X - 2z_n X^\top Q(S)X]\}G(S)n^{-1/2}. \quad\text{(A.10)}
\end{aligned}
$$

Next we will show that the trace of $\partial_\alpha G(S)$ is bounded by $n^{-1/2}$. By the inequality $\|AB\| \leq \|A\|\|B\|$ and the lemma A.7 below, we only need to show that the sum of the absolute values of the eigenvalues of $Q(S)$ and the spectral norms of

$$
X^\top X, \quad S^\top S, \quad P(S), \quad (P(S) - z_n I_p)^{-2}, \quad G(S)
$$

are all bounded above by some constants only dependent on $c$ and $\xi$.

**Lemma A.7.** *(Trace of products). Suppose $A, B$ are two $n \times n$ diagonalizable complex matrices, then*

$$
|\operatorname{tr}(AB)| \leq |\lambda|_{\max}(A)\sum_{i=1}^n |\mu_i|,
$$

*where $|\lambda|_{\max}(A)$ is the largest absolute value of eigenvalues of $A$ and $\mu_i$ are the eigenvalues of $B$.*

Note that

$$
Q(S) = E_{ji}S + S^\top E_{ij} = e_j S_{i\cdot} + S_{i\cdot}^\top e_j^\top,
$$

where $e_j$ is an $n \times 1$ vector with the $j$th entry equal to 1 and the rest equal to 0, $S_{i\cdot}$ is the $i$th row of $S$. The eigenvalues of $Q(S)$ are $S_{ij} \pm \|S_{i\cdot}\|$, according to Lemma A.8 below.

**Lemma A.8.** *(Rank two matrices.) Let $u, v \in \mathbb{R}^n$ and $u^\top v \neq 0$, then the nonzero eigenvalues of $uv^\top + vu^\top$ are $u^\top v \pm \|u\|\|v\|$, both with multiplicity 1.*

First note that $|S_{ij}| \leq \sigma_{\max}(S)$ and $\|S_{i\cdot}\| \xrightarrow{a.s.} 1$ by the law of large number. It is also known that as $n \to \infty$ and $r/n \to \xi$, we have

$$
\lambda_{\min}(S^\top S) \xrightarrow{a.s.} (1 - \sqrt{\xi})^2, \quad \lambda_{\max}(S^\top S) \xrightarrow{a.s.} (1 + \sqrt{\xi})^2,
$$

see Bai and Silverstein [2010]. So the sum of the absolute values of the eigenvalues of $Q(S)$ is bounded above by $2(2 + \sqrt{\xi})$, almost surely as $n$ tends to infinity.

By our assumption, the eigenvalues of $X^\top X$ are bounded in the interval $[\frac{1}{c^2}, c^2]$.

Suppose the eigenvalues of $X^\top S^\top S X$ are $\lambda_1 \geq \ldots \geq \lambda_p$. So almost surely,

$$
\begin{aligned}
\lambda_p &\geq \lambda_{\min}(X^\top X)\lambda_{\min}(S^\top S) \geq \frac{1}{c^2}(1 - \sqrt{\xi})^2, \\
\lambda_1 &\leq \lambda_{\max}(X^\top X)\lambda_{\max}(S^\top S) \leq c^2(1 + \sqrt{\xi})^2,
\end{aligned}
$$

Since the complex matrix $X^\top S^\top S X - z_n I_p$ is diagonalizable, and its eigenvalues are $\lambda_1 - z_n, \ldots, \lambda_p - z_n$. Thus the eigenvalues of $(X^\top S^\top S X - z_n I_p)^{-2}$ are $\frac{1}{(\lambda_1 - z_n)^2}, \ldots, \frac{1}{(\lambda_p - z_n)^2}$. Because $\lambda_i \in \mathbb{R}$, $z_n = i/n$ and $|\lambda_i - z_n| > |\lambda_i|$, the largest absolute eigenvalue of $(X^\top S^\top S X - z_n I_p)^{-2}$ is bounded above by $\frac{1}{\lambda_p^2}$, that is, $\|(P(S) - z_n I_p)^{-2}\| \leq \frac{1}{\lambda_p^2} \leq \frac{c^4}{(1 - \sqrt{\xi})^4}$.

We also have

$$\|G(S)\| \leq \|(P(S) - z_n I_p)^{-2}\| \|X^\top (S^\top S)^2 X\|$$

$$\leq \frac{c^4}{(1 - \sqrt{\xi})^4} c^2 (1 + \sqrt{\xi})^4 = c^6 \frac{(1 + \sqrt{\xi})^4}{(1 - \sqrt{\xi})^4}.$$

Thus $\mathrm{tr}[\partial_\alpha G(S)]$ is bounded by $O(n^{-1/2})$. Since $p/n \to \gamma$, there exists a constant $\phi_1(c, \gamma, \xi)$, such that

$$|f_N| = \frac{1}{p} |\mathrm{tr}[\partial_\alpha G(S)]| \leq \phi_1(c, \gamma, \xi) n^{-3/2}.$$

Next we will bound the second derivative of $f_N$ from above by $n^{-2}$. Take the second derivative w.r.t. to $\alpha$ on both sides of (A.6), we have

$$\partial_\alpha^2 [(X^\top S^\top S X - z_n I_p)^2] \cdot G(S) + 2\partial_\alpha [(X^\top S^\top S X - z_n I_p)^2] \cdot \partial_\alpha G(S) + (X^\top S^\top S X - z_n I_p)^2 \partial_\alpha^2 G(S)$$
$$= \partial_\alpha^2 [X^\top (S^\top S)^2 X], \tag{A.11}$$

and thus

$$\partial_\alpha^2 G(S) = (X^\top S^\top S X - z_n I_p)^{-2} [\partial_\alpha^2 [X^\top (S^\top S)^2 X] - \partial_\alpha^2 [(X^\top S^\top S X - z_n I_p)^2] \cdot G(S) -$$
$$2\partial_\alpha [(X^\top S^\top S X - z_n I_p)^2] \cdot \partial_\alpha G(S)]. \tag{A.12}$$

Using (A.8), we have

$$\partial_\alpha^2 [(X^\top S^\top S X - z_n I_p)^2] = \partial_\alpha [X^\top (E_{ji} S + S^\top E_{ij}) X X^\top S^\top S X + X^\top S^\top S X X^\top (E_{ji} S + S^\top E_{ij}) X$$
$$- 2 z_n X^\top (E_{ji} S + S^\top E_{ij}) X] n^{-1/2}$$
$$= \{X^\top (E_{ji} E_{ij} + E_{ji} E_{ij}) X X^\top S^\top S X +$$
$$X^\top (E_{ji} S + S^\top E_{ij}) X X^\top (E_{ji} S + S^\top E_{ij}) X +$$
$$X^\top (E_{ji} S + S^\top E_{ij}) X X^\top (E_{ji} S + S^\top E_{ij}) X +$$
$$X^\top S^\top S X X^\top (E_{ji} E_{ij} + E_{ji} E_{ij}) X -$$
$$2 z_n X^\top (E_{ji} E_{ij} + E_{ji} E_{ij}) X\} \frac{1}{n}$$
$$= \{2 (X^\top (E_{ji} S + S^\top E_{ij}) X)^2$$
$$+ 2 X^\top E_{jj} X X^\top S^\top S X + 2 X^\top S^\top S X X^\top E_{jj} X - 4 z_n X^\top E_{jj} X\} \frac{1}{n}.$$

Using (A.9), we have

$$\partial_\alpha^2 [X^\top (S^\top S)^2 X] = \partial_\alpha [\{X^\top [(E_{ji} S + S^\top E_{ij})(S^\top S) + (S^\top S)(E_{ji} S + S^\top E_{ij})] X\}] n^{-1/2}$$
$$= X^\top [2 E_{jj} S^\top S + 2 (E_{ji} S + S^\top E_{ij})^2 + 2 S^\top S E_{jj}] X \frac{1}{n}.$$

By the same arguments, we can show that the traces of the three terms on the right hand side of (A.12) are bounded above by $n^{-1}$ in magnitude, therefore the second derivative of $f_N$ is bounded by $n^{-2}$. Also by the same reasoning, we can show that there exists some constant $\phi_3(c, \xi, \gamma)$, such that $|\partial_\alpha^3 f_N(s)| \leq \phi_3(c, \xi, \gamma) N^{-5/4}$, holds almost surely as $n$ goes to infinity.

We then use similar methods to bound the third derivative of $g_N(s)$. Define

$$H_n(S) = (X^\top S^\top SX - z_n I_p)^{-1} X^\top X (X^\top S^\top SX - z_n I_p)^{-1} X^\top (S^\top S)^2 X,$$

then

$$g_N(s) = \frac{1}{p} \operatorname{tr}[H_n(s)].$$

Note also that

$$(X^\top S^\top SX - z_n I_p)(X^\top X)^{-1}(X^\top S^\top SX - z_n I_p)H_n(S) = X^\top (S^\top S)^2 X.$$

Taking derivative w.r.t. to $\alpha$ on both sides we have

$$n^{-1/2}[X^\top (E_{ji} S + S^\top E_{ij})X(X^\top X)^{-1}(X^\top S^\top SX - z_n I_p)H_n(S)+$$
$$(X^\top S^\top SX - z_n I_p)(X^\top X)^{-1}X^\top (E_{ji} S + S^\top E_{ij})X H_n(S)]+$$
$$(X^\top S^\top SX - z_n I_p)(X^\top X)^{-1}(X^\top S^\top SX - z_n I_p)\partial_\alpha H_n(S)$$
$$= n^{-1/2}[X^\top (E_{ji} S + S^\top E_{ij})S^\top SX + X^\top S^\top S(E_{ji} S + S^\top E_{ij})X].$$

Using similar techniques, we can show that almost surely $\frac{1}{p}|\operatorname{tr}[\partial_\alpha H_n(S)]|$ is bounded in magnitude by $n^{-3/2}$, $\frac{1}{p}|\operatorname{tr}[\partial_\alpha^2 H_n(S)]|$ is bounded in magnitude by $n^{-2}$, and $\frac{1}{p}|\operatorname{tr}[\partial_\alpha^3 H_n(S)]|$ is bounded in magnitude by $n^{-5/2}$. Therefore almost surely $|\partial_\alpha^3 g_N(s)| \leq \phi_3' N^{-5/4}$, for some $\phi_3' = \phi_3'(c, \xi, \gamma)$. Take $\phi = \max(\phi_3, \phi_3')$, and the proof of Lemma A.6 is done. □

*Proof.* (Proof of Lemma A.7) Consider the eigendecompositions of $A, B$,

$$A = Q \begin{pmatrix} \lambda_1 & & \\ & \ddots & \\ & & \lambda_n \end{pmatrix} Q^\top, \ B = P \begin{pmatrix} \mu_1 & & \\ & \ddots & \\ & & \mu_n \end{pmatrix} P^\top,$$

then

$$\operatorname{tr}(AB) = \operatorname{tr}(Q \begin{pmatrix} \lambda_1 & & \\ & \ddots & \\ & & \lambda_n \end{pmatrix} Q^\top P \begin{pmatrix} \mu_1 & & \\ & \ddots & \\ & & \mu_n \end{pmatrix} P^\top).$$

Denote the $n$ columns of $Q^\top P$ as $v_1, \ldots, v_n$, which are orthonormal. Then

$$|\operatorname{tr}(AB)| = |\operatorname{tr}(\begin{pmatrix} \lambda_1 & & \\ & \ddots & \\ & & \lambda_n \end{pmatrix} \sum_{i=1}^n \mu_i v_i v_i^\top)|$$

$$= |\sum_{i=1}^n \mu_i v_i^\top \begin{pmatrix} \lambda_1 & & \\ & \ddots & \\ & & \lambda_n \end{pmatrix} v_i| \leq \sum_{i=1}^n |\mu_i||\lambda|_{\max}(A).$$

This finishes the proof. □

*Proof.* (Proof of Lemma A.8) It is easy to see that $uv^\top + vu^\top$ has rank 2 and

$$(uv^\top + vu^\top)(\frac{u}{\|u\|} + \frac{v}{\|v\|}) = (u^\top v + \|u\|\|v\|)(\frac{u}{\|u\|} + \frac{v}{\|v\|}),$$
$$(uv^\top + vu^\top)(\frac{u}{\|u\|} - \frac{v}{\|v\|}) = (u^\top v - \|u\|\|v\|)(\frac{u}{\|u\|} - \frac{v}{\|v\|}).$$

This finishes the proof. □

### A.6.2 Proof of Lemma A.4

Let $A = X^\top S^\top SX$ and $B = X^\top S^\top SX - z_n I_n$, and note that we have the relationship

$$A^{-2} - B^{-2} = B^{-1}(B - A)A^{-2} + B^{-2}(B - A)A^{-1} = -z_n(B^{-1}A^{-2} + B^{-2}A^{-1}).$$

Thus

$$f_N(s) - f_\infty(t) = \frac{1}{p}\operatorname{tr}[(A^{-2} - B^{-2})X^\top(S^\top S)^2 X]$$

$$= -z_n\frac{1}{p}\operatorname{tr}[(B^{-1}A^{-2} + B^{-2}A^{-1})X^\top(S^\top S)^2 X].$$

If the eigenvalues of $A$ are $\lambda_1 \geq \ldots \geq \lambda_p > 0$, then the eigenvalues of $B$ are $\lambda_1 - z_n, \ldots, \lambda_p - z_n$. By Lemma A.7, we have

$$\frac{1}{p}|\operatorname{tr}[B^{-1}A^{-2}X^\top(S^\top S)^2 X]| \leq \|A^{-2}X^\top(S^\top S)^2 X\|\frac{1}{p}\sum_{i=1}^p \frac{1}{|\lambda_i - z_n|}$$

$$\leq \frac{1}{\lambda_p^2}\|X^\top X\|\|S^\top S\|^2\frac{1}{\lambda_p}.$$

Recall that $\lambda_p \geq \frac{1}{c^2}(1 - \sqrt{\xi})^2$, then we have

$$\frac{1}{p}|\operatorname{tr}[B^{-1}A^{-2}X^\top(S^\top S)^2 X]| \leq c^8\frac{(1 + \sqrt{\xi})^4}{(1 - \sqrt{\xi})^6}.$$

By the same argument, we have

$$\frac{1}{p}|\operatorname{tr}[B^{-2}A^{-1}X^\top(S^\top S)^2 X]| \leq c^8\frac{(1 + \sqrt{\xi})^4}{(1 - \sqrt{\xi})^6}.$$

Hence

$$|f_N(s) - f_\infty(s)| \leq \frac{1}{p}2c^8\frac{(1 + \sqrt{\xi})^4}{(1 - \sqrt{\xi})^6}$$

holds almost surely. Hence, $f_N(s) - f_\infty(s) \xrightarrow{a.s.} 0$. By the bounded convergence theorem, we have $\lim_{n\to\infty}|\mathbb{E}[f_N(s)] - \mathbb{E}[f_\infty(s)]| = 0$. The other three limit statements can be proved similarly. This finishes the proof.

## A.7  Proof of Theorem 2.3

Suppose that $X$ has the SVD factorization $X = U\Lambda V^\top$ and let $S_1 = SU$. The majority of the proof will deal with the following quantities:

$$\operatorname{tr}[(X^\top X)^{-1}] = \operatorname{tr}(\Lambda^{-2}),$$
$$\operatorname{tr}[(X^\top S^\top SX)^{-1}] = \operatorname{tr}[(\Lambda U^\top S^\top SU\Lambda)^{-1}] = \operatorname{tr}[(\Lambda S_1^\top S_1 \Lambda)^{-1}],$$
$$\operatorname{tr}[(X^\top S^\top SX)^{-1} X^\top X] = \operatorname{tr}[(U^\top S^\top SU)^{-1}] = \operatorname{tr}[(S_1^\top S_1)^{-1}].$$

Since we are finding the limits of these quantities, we add the subscript $n$ to matrices like $S_n, U_n$ from now on. Since both $S_n$ and $U_n$ are rectangular orthogonal matrices, we embed them into full orthogonal matrices as

$$\mathbb{S}_n = \begin{pmatrix} S_n \\ S_n^\perp \end{pmatrix}, \mathbb{U}_n = \begin{pmatrix} U_n \\ U_n^\perp \end{pmatrix}.$$

Suppose $\frac{1}{p}\Lambda_n S_{1,n}^\top S_{1,n}\Lambda_n$ has an l.s.d. bounded away from zero. Then, the limit of $\frac{1}{p}\operatorname{tr}[(\frac{1}{p}\Lambda_n S_{1,n}^\top S_{1,n}\Lambda_n)^{-1}]$ must equal to the Stieltjes transform of its l.s.d. evaluated at zero. Therefore, we first find the Stieltjes transforms of the l.s.d. of the matrices $\frac{1}{p}\Lambda_n S_{1,n}^\top S_{1,n}\Lambda_n$. The same applies to $\operatorname{tr}[(S_{1,n}^\top S_{1,n})^{-1}]$, except that we replace $\Lambda_n$ with the identity matrix.

Since $\Lambda_n S_{1,n}^\top S_{1,n}\Lambda_n$ and $S_{1,n}\Lambda_n^2 S_{1,n}^\top$ have the same non-zero eigenvalues, we first find the l.s.d. of $\frac{1}{n}S_{1,n}\Lambda_n^2 S_{1,n}^\top$. Note that

$$S_{1,n} = S_n U_n = \begin{pmatrix} I_r & 0 \end{pmatrix} \begin{pmatrix} S_n \\ S_n^\perp \end{pmatrix} \begin{pmatrix} U_n & U_n^\perp \end{pmatrix} \begin{pmatrix} I_p \\ 0 \end{pmatrix}$$

$$= \begin{pmatrix} I_r & 0 \end{pmatrix} \mathbb{S}_n \mathbb{U}_n \begin{pmatrix} I_p \\ 0 \end{pmatrix}.$$

Let $\mathbb{W}_n = \mathbb{S}_n \mathbb{U}_n$, which is again an $n \times n$ Haar-distributed matrix due to the orthogonal invariance of the Haar distribution. Then

$$S_{1,n}\Lambda_n^2 S_{1,n}^\top = \begin{pmatrix} I_r & 0 \end{pmatrix} \mathbb{W}_n \begin{pmatrix} I_p \\ 0 \end{pmatrix} \Lambda_n^2 \begin{pmatrix} I_p & 0 \end{pmatrix} \mathbb{W}_n^\top \begin{pmatrix} I_r \\ 0 \end{pmatrix}.$$

Define

$$C_n = \frac{1}{n} \begin{pmatrix} I_r & 0 \\ 0 & 0 \end{pmatrix} \mathbb{W}_n \begin{pmatrix} \Lambda_n^2 & 0 \\ 0 & 0 \end{pmatrix} \mathbb{W}_n^\top \begin{pmatrix} I_r & 0 \\ 0 & 0 \end{pmatrix} = \frac{1}{n} \begin{pmatrix} S_{1,n}\Lambda_n^2 S_{1,n}^\top & 0 \\ 0 & 0 \end{pmatrix}. \tag{A.13}$$

Since $X$ has an l.s.d., we get that the e.s.d. of $\begin{pmatrix} \Lambda_n^2 & 0 \\ 0 & 0 \end{pmatrix}$ converges to some fixed distribution $F_\Lambda$, and we know that the e.s.d. of $\begin{pmatrix} I_r & 0 \\ 0 & 0 \end{pmatrix}$ converges to $F_\xi = \xi\delta_1 + (1-\xi)\delta_0$. Then according to Hachem [2008] or Theorem 4.11 of Couillet and Debbah [2011], the e.s.d. of $C_n$ converges to a distribution $F_C$, whose $\eta$-transform $\eta_C$ is the unique solution of the following system of equations,

defined for all $z \in \mathbb{C}^+$:

$$\eta_C(z) = \int \frac{1}{z\gamma(z)t+1}dF_\xi(t) = \frac{\xi}{z\gamma(z)+1} + (1-\xi),$$

$$\gamma(z) = \int \frac{t}{\eta_C(z)+z\delta(z)t}dF_\Lambda(t),$$

$$\delta(z) = \int \frac{t}{z\gamma(z)t+1}dF_\xi(t) = \frac{\xi}{z\gamma(z)+1}.$$

Moreover, we note that if the support of $F_\Lambda$ outside of the point mass at zero is bounded away from the origin, then the same is also true for $F_C$. Indeed, this follows directly from the form of $\Lambda_n S_{1,n}^\top S_{1,n} \Lambda_n$, as its smallest eigenvalue can be bounded below as

$$\lambda_{\min}(\Lambda_n S_{1,n}^\top S_{1,n} \Lambda_n) \geq \lambda_{\min}(\Lambda_n)^2 \lambda_{\min}(S_{1,n}^\top S_{1,n}).$$

Moreover, by assumption $\lambda_{\min}(\Lambda_n) > c > 0$ for some universal constant $c$, and clearly $\lambda_{\min}(S_{1,n}^\top S_{1,n}) = 1$, as $S_{1,n}$ is a partial orthogonal matrix. This ensures that we can use the Stieltjes transform as a tool to calculate the limiting traces of the inverse.

Returning to our equations, using the first and the third equations to solve for $\delta(z)$ and $\gamma(z)$ in terms of $\eta_C(z)$, substituting them in the second equation, we get the following fixed point equation

$$\eta_C(z) = \eta_\Lambda(z(1+\frac{\xi-1}{\eta_C(z)})). \tag{A.14}$$

According to the definition of $\eta$-transform (A.1), for any distribution $F$ with a point mass $f_F(0)$ at zero, we have

$$\eta_F(z) = \int_{t\neq0} \frac{1}{1+zt}dF(t) + f_F(0).$$

Note that $f_C(0) = f_\Lambda(0) = 1-\gamma$. Since the l.s.d. of $X$ is compactly supported and bounded away from the origin, we know $\inf[supp(f_\Lambda) \cap \mathbb{R}^*]$ and $\inf[supp(f_\Lambda) \cap \mathbb{R}^*]$ are greater than zero, thus $\frac{1}{t}$ is integrable on the set $\{t>0\}$ w.r.t. $F_\Lambda$ and $F_C$. Since $|\frac{z}{1+tz}| < \frac{1}{t}$ when $z>0, t>0$, by the dominated convergence theorem we have

$$\lim_{z\to\infty} \int_{t\neq0} \frac{z}{1+tz}dF_C(t) = \int_{t\neq0} \frac{1}{t}dF_C(t),$$

$$\lim_{z\to\infty} \int_{t\neq0} \frac{z}{1+tz}dF_\Lambda(t) = \int_{t\neq0} \frac{1}{t}dF_\Lambda(t),$$

and hence

$$\int_{t\neq0} \frac{1}{t}dF_C(t) = \lim_{z\to\infty} z(\eta_C(z)-(1-\gamma)), \tag{A.15}$$

$$\int_{t\neq0} \frac{1}{t}dF_\Lambda(t) = \lim_{z\to\infty} z(\eta_\Lambda(z)-(1-\gamma)), \tag{A.16}$$

and

$$\lim_{z\to\infty} \eta_C(z) = \lim_{z\to\infty} \int_{t\neq 0} \frac{1}{1+zt} dF_C(t) + (1-\gamma)$$

$$= \int_{t\neq 0} \lim_{z\to\infty} \frac{1}{1+zt} dF_C(t) + (1-\gamma)$$

$$= 1 - \gamma. \tag{A.17}$$

Subtracting $1 - \gamma$ from both sides of (A.14), multiplying by $z(1 + \frac{\xi-1}{\eta_C(z)})$, letting $z \to \infty$, we obtain

$$\lim_{z\to\infty} z(1 + \frac{\xi-1}{\eta_C(z)})[\eta_C(z) - (1-\gamma)] = \lim_{z\to\infty} z(1 + \frac{\xi-1}{\eta_C(z)})[\eta_\Lambda(z(1 + \frac{\xi-1}{\eta_C(z)})) - (1-\gamma)].$$

Note that RHS equals $\int_{t\neq 0} \frac{1}{t} dF_\Lambda(t)$ by (A.16), and

$$LHS = \lim_{z\to\infty} z(1 + \frac{\xi-1}{\eta_C(z)})[\eta_C(z) - (1-\gamma)]$$

$$= \lim_{z\to\infty} z[\eta_C(z) - (1-\gamma)](1 + \frac{\xi-1}{1-\gamma})$$

$$= \int_{t\neq 0} \frac{1}{t} dF_C(t) \frac{\xi-\gamma}{1-\gamma},$$

where the second and the third equations follow from (A.17) and (A.16). This shows that

$$\int_{t\neq 0} \frac{1}{t} dF_\Lambda(t) = \frac{\xi-\gamma}{1-\gamma} \int_{t\neq 0} \frac{1}{t} dF_C(t),$$

therefore we have proved that as $n \to \infty$,

$$\frac{\text{tr}[(\Lambda S_1^\top S_1^\top \Lambda)^{-1}]}{\text{tr}(\Lambda^{-2})} \to \frac{\int_{t\neq 0} \frac{1}{t} dF_C(t)}{\int_{t\neq 0} \frac{1}{t} dF_\Lambda(t),} = \frac{1-\gamma}{\xi-\gamma},$$

thus

$$\lim_{n\to\infty} VE(\hat{\beta}_s, \hat{\beta}) = \frac{1-\gamma}{\xi-\gamma}.$$

This finishes the evaluation of $VE$.

Next, to evaluate of $PE$, we argue as follows: In the definition of $C_n$ in (A.13), replace $\Lambda_n$ by the identity matrix. Since the results do not depend the l.s.d. of $\Lambda_n$, it follows directly that

$$PE = \frac{\text{tr}[(X^\top S^\top SX)^{-1} X^\top X]}{p} = \frac{\text{tr}[(S_1^\top S_1)^{-1}]}{\text{tr}(I_p)} \to \frac{1-\gamma}{\xi-\gamma}.$$

Next, to evaluate the limit of $OE$, we use the additional assumption on $X$, that is, $X = Z\Sigma^{1/2}$, where $Z$ has iid entries of zero mean, unit variance and finite fourth moment.

Note that (with convergence below always meaning almost sure convergence)

$$\mathbb{E}\left[x_t^\top (X^\top X)^{-1} x_t)\right] \to \frac{\gamma}{1-\gamma},$$

which has been proved in Section A.3, and

$$1 + \mathbb{E}\left[x_t^\top (X^\top X)^{-1} x_t\right] \to 1 + \frac{\gamma}{1-\gamma} = \frac{1}{1-\gamma}.$$

On the other hand,

$$\mathbb{E}\left[x_t^\top (X^\top S^\top S X)^{-1} x_t\right] = \mathrm{tr}(\mathbb{E}\left[X^\top S^\top S X\right]^{-1} \mathbb{E}\left[x_t x_t^\top\right])$$
$$= \mathrm{tr}(\mathbb{E}\left[(\Sigma^{1/2} Z^\top S^\top S Z \Sigma^{1/2})^{-1}\right] \Sigma) = \mathrm{tr}(\mathbb{E}\left[Z^\top S^\top S Z\right]^{-1}).$$

Define $C_n = \frac{1}{n} Z^\top S^\top S Z$, then the e.s.d. of $C_n$ converges to a distribution $F_C$, whose Stieltjes transform $m(z) = m_C(z), z \in \mathbb{C}^+$ is given by [Bai and Silverstein, 2010]

$$m(z) = \frac{1}{\int \frac{s}{1+\gamma s e} dF_{S^\top S}(s) - z} = \frac{1}{\frac{\xi}{1+\gamma e} - z},$$

where

$$e = \frac{1}{\int \frac{s}{1+\gamma s e} dF_{S^\top S}(s) - z} = \frac{1}{\frac{\xi}{1+\gamma e} - z}.$$

And here $F_{S^\top S}$ is the l.s.d. of $S^\top S$, which is $\xi \delta_1 + (1-\xi)\delta_0$. Solving these equations gives

$$m(z) = e(z) = \frac{\xi - \gamma - z + \sqrt{(\xi - \gamma - z)^2 - 4z\gamma}}{2z\gamma}.$$

Therefore

$$\lim_{z \to 0} m(z) = \frac{-1 - \frac{2(\gamma - \xi) - 4\gamma}{2(\xi - \gamma)}}{2\gamma} = \frac{-1 + \frac{\xi + \gamma}{\xi - \gamma}}{2\gamma} = \frac{1}{\xi - \gamma}.$$

Thus

$$\mathrm{tr}((Z^\top S^\top S Z)^{-1}) = \frac{1}{n} \mathrm{tr}((\frac{1}{n} Z^\top S^\top S Z)^{-1}) \xrightarrow{a.s.} \gamma m_C(0) = \frac{\gamma}{\xi - \gamma}.$$

Therefore

$$1 + \mathbb{E}\left[x_t^\top (X^\top S^\top S X)^{-1} x_t\right] \to 1 + \frac{\gamma}{\xi - \gamma} = \frac{1}{1 - \gamma/\xi},$$

and we have proved

$$\lim_{n \to \infty} OE(\hat{\beta}_s, \hat{\beta}) = \lim_{n \to \infty} \frac{1 + \mathbb{E}\left[x_t^\top (X^\top S^\top S X)^{-1} x_t\right]}{1 + \mathbb{E}\left[x_t^\top (X^\top X)^{-1} x_t\right]} = \frac{1 - \gamma}{1 - \gamma/\xi}.$$

This finishes the proof.

### A.7.1 Checking the free multiplicative convolution property

Recall that the $S$-transform of a distribution $F$ is defined as the solution to the equation

$$m_F(\frac{z+1}{zS(z)}) = -zS(z).$$

For more references, see for instance Voiculescu et al. [1992], Hiai and Petz [2006], Nica and Speicher [2006], Anderson et al. [2010].

Since $m(\frac{z+1}{zS(z)}) = -zS(z), \eta(z) = \frac{1}{z}m(-\frac{1}{z})$, we have

$$-zS(z) = m(\frac{z+1}{zS(z)}) = -\frac{zS(z)}{z+1}\eta(-\frac{zS(z)}{z+1}),$$

where $S(z)$ is the $S$-transform. Therefore

$$\eta_\Lambda(-\frac{zS_\Lambda(z)}{z+1}) = z+1,\ \eta_C(-\frac{zS_C(z)}{z+1}) = z+1.$$

Let $x = -\frac{z}{z+1}S_C(z)$, then $\eta_C(x) = z+1$ and (A.14) gives

$$z+1 = \eta_C(x) = \eta_\Lambda(x(1+\frac{\xi-1}{\eta_C(x)})) = \eta_\Lambda(-\frac{z}{z+1}S_C(z)(1+\frac{\xi-1}{z+1}))$$

$$= \eta_\Lambda(-\frac{z}{z+1}S_C(z)\frac{z+\xi}{z+1}) = \eta_\Lambda(-\frac{z}{z+1}S_\Lambda(z)).$$

Therefore $S_\Lambda = \frac{z+\xi}{z+1}S_C(z)$, and equivalently $S_C(z) = S_\Lambda(z)\frac{z+1}{z+\xi}$. Let $S_0(z) = \frac{z+1}{z+\xi}$ be the $S$-transform of some distribution $F_0$, then the corresponding Stieltjes transform is $m_0(z) = \frac{\xi}{1-z} + \frac{1-\xi}{-z}$, which is the Stieltjes transform for $F_0 = \xi\delta_1 + (1-\xi)\delta_0$. This shows that $F_C$ is a freely multiplicative convolution of $F_\Lambda$ and $\xi\delta_1 + (1-\xi)\delta_0$.

## A.8 Proof of Theorem 2.4

Note that $B, H$ and $D$ are all symmetric matrices satisfying

$$B^2 = B,\ H^2 = I_n,\ D^2 = I_n,$$

and $P$ is also an orthogonal matrix, therefore

$$S^\top S = P^\top DHBHDP$$

$$(S^\top S)^2 = P^\top DHBHDPP^\top DHBHDP$$

$$= P^\top DHBHDP = S^\top S.$$

By Proposition A.2, we only need to find

$$\mathrm{tr}[(X^\top S^\top SX)^{-1}] = \mathrm{tr}[(X^\top P^\top DHBHDPX)^{-1}], \tag{A.18}$$

and

$$\mathrm{tr}[(X^\top S^\top SX)^{-1}X^\top X] = \mathrm{tr}[(X^\top P^\top DHBHDPX)^{-1}X^\top X]. \tag{A.19}$$

We first have the following observation.

**Lemma A.9.** *For a uniformly distributed permutation matrix $P$, diagonal matrix $B$ with iid diagonal entries of distribution $\mu_B = \frac{r}{n}\delta_1 + (1 - \frac{r}{n})\delta_0$, diagonal matrix $D$ with iid sign random variables, equal to $\pm 1$ with probability one half, and Hadamard matrix $H$, we have the following equation in distribution*

$$X^\top (P^\top DH)B(HDP)X \overset{d}{=} X^\top (P^\top DHDP)B(P^\top DHDP)X.$$

This is true, because we are simply permuting the diagonal matrix of iid Bernoullis in the middle term; but see the end of this section for a formal proof. We call $DP$ the signed-permutation matrix and $W = P^\top DHDP$ the bi-signed-permutation Hadamard matrix. Thus by equations (A.18), (A.19), and Lemma A.9,

$$\mathbb{E}\left[\text{tr}[(X^\top S^\top SX)^{-1}]\right] = \mathbb{E}\left[\text{tr}[(X^\top (P^\top DHDP)B(P^\top DHDP)X)^{-1}]\right]$$
$$= \mathbb{E}\left[\text{tr}[(X^\top WBWX)^{-1}]\right],$$
$$\mathbb{E}\left[\text{tr}[(X^\top S^\top SX)^{-1}X^\top X]\right] = \mathbb{E}\left[\text{tr}[(X^\top (P^\top DHDP)B(P^\top DHDP)X)^{-1}X^\top X]\right]$$
$$= \mathbb{E}\left[\text{tr}[(X^\top WBWX)^{-1}X^\top X]\right].$$

Since $X^\top WBWX$ has the same nonzero eigenvalues as $BWXX^\top WB$, we first find the l.s.d. of

$$C_n = \frac{1}{n}B_n W_n X_n X_n^\top W_n B_n.$$

The following lemma states the asymptotic freeness regarding Hadamard matrix, which will be used to find the l.s.d. of $C_n$. For more references on free probability, see for instance Voiculescu et al. [1992], Hiai and Petz [2006], Nica and Speicher [2006], Anderson et al. [2010].

**Lemma A.10.** *(Freeness of bi-signed-permutation Hadamard matrix) Let $X_n, B_n, W_n$ be defined above, that is, $X_n$ is an $n \times n$ deterministic matrix with uniformly bounded spectral norm and has l.s.d. $\mu_X$, $B_n$ is a diagonal matrix with iid diagonal entries, and $W_n$ is a bi-signed-permutation matrix. Then*

$$\{B_n, \frac{1}{n}W_n X_n X_n^\top W_n\}$$

*are asymptotically free in the limit of the non-commutative probability spaces of random matrices, as described in Section A.1. The law of*

$$C_n = \frac{1}{n}B_n W_n X_n X_n^\top W_n B_n$$

*converges to the freely multiplicative convolution of $\mu_B$ and $\mu_X$, that is, $C_n$ has l.s.d. $\mu_C = \mu_B \boxtimes \mu_X$.*

This follows directly from Corollaries 3.5, 3.7 of Anderson and Farrell [2014]. See also Lemma 1 of Tulino et al. [2010] for earlier results on the Fourier transform.

We use $\mu_B$ and $\mu_X$ to denote the elements in the limiting non-commutative probability space, their laws, and their corresponding probability distributions interchangeably. Since $\mu_B = \xi\delta_1 + (1 - \xi)\delta_0$, we have $S_{\mu_B} = \frac{z+1}{z+\xi}$. From the aymptotic freeness, it follows that the $S$-transform of $\mu_C$ is the product of that of $\mu_B, \mu_X$, so that

$$S_{\mu_C}(z) = S_{\mu_X}(z)S_{\mu_B}(z) = S_{\mu_X}(z)\frac{z+1}{z+\xi}.$$

We will now simplify this relation. First, note that by the definition of the S-transform, we have

$$\eta_{\mu_C}(-\frac{z}{z+1}S_{\mu_C}(z)) = z+1.$$

Letting $y = -\frac{z}{z+1}S_{\mu_C}$, we have $\eta_{\mu_C}(y) = z+1$. In addition, we can simplify the original relation as

$$S_{\mu_X} = \frac{z+\xi}{z+1}S_{\mu_C}(z) = -\frac{z+\xi}{z}y,$$

$$z+1 = \eta_{\mu_X}(-\frac{z}{z+1}S_{\mu_X}(z)) = \eta_{\mu_X}(\frac{z+\xi}{z+1}y)$$

$$= \eta_{\mu_X}((1+\frac{\xi-1}{z+1})y) = \eta_{\mu_X}((1+\frac{\xi-1}{\eta_{\mu_C}(y)})y) = \eta_{\mu_C}(z).$$

So we have obtained

$$\eta_{\mu_X}((1+\frac{\xi-1}{\eta_{\mu_C}(y)})y) = \eta_{\mu_C}(y).$$

This is the same equation as what we obtained in (A.14) in the proof of Haar projection. Therefore as $n \to \infty$, we have as required

$$\lim_{n\to\infty} VE(\hat{\beta}_s, \hat{\beta}) = \frac{1-\gamma}{\xi-\gamma}.$$

Next we consider

$$\mathbb{E}\left[\text{tr}[(X^\top W B W X)^{-1}X^\top X]\right].$$

Since $X$ has the SVD $X = U\Lambda V^\top$, we have

$$\mathbb{E}\left[\text{tr}[(X^\top W B W X)^{-1}X^\top X]\right] = \mathbb{E}\left[\text{tr}[(U^\top W B W U)^{-1}]\right].$$

Thus we can repeat the above reasoning, except that we replace $X$ by $U$. Since the result does not depend on $X$, we have

$$\lim_{n\to\infty} PE(\hat{\beta}_s, \hat{\beta}) = \lim_{n\to\infty} \frac{\mathbb{E}\left[\text{tr}[(X^\top S^\top S X)^{-1}X^\top X]\right]}{p}$$

$$= \lim_{n\to\infty} \frac{\mathbb{E}\left[\text{tr}[(U^\top W B W U)^{-1}]\right]}{\text{tr}[U^\top U]}$$

$$= \lim_{n\to\infty} VE(\hat{\beta}_s, \hat{\beta}) = \frac{1-\gamma}{\xi-\gamma}.$$

For $OE$, since $S$ satisfies $(S^\top S)^2 = S^\top S$ and the e.s.d. of $S^\top S$ converges to $\xi\delta_1 + (1-\xi)\delta_0$, the same reasoning as in Theorem 2.3 also holds in this case for Hadamard projection. This finishes the proof.

*Proof.* (Proof of Lemma A.9) Note that both $B$ and $D$ are diagonal matrices whose diagonal entries are iid random variables, and $P$ is a permutation matrix. Define $\tilde{B} = PBP^\top$ and $\tilde{D} = P^\top DP$, then we have

$$\tilde{B} \stackrel{d}{=} B, \quad \tilde{D} \stackrel{d}{=} D$$

and

$$DP = P\tilde{D}, \quad P^\top D = \tilde{D}P^\top. \tag{A.20}$$

Hence

$$
\begin{aligned}
X^\top P^\top DHDPBP^\top DHDPX &= X^\top P^\top DHP\tilde{D}B\tilde{D}P^\top HDPX \\
&= X^\top P^\top DHPB\tilde{D}^2 P^\top HDPX \\
&= X^\top P^\top DHPBP^\top HDPX \\
&= X^\top P^\top DH\tilde{B}HDPX \\
&\overset{d}{=} X^\top P^\top DHBHDPX,
\end{aligned}
$$

where the first equation follows from (A.20), the second equation holds because $\tilde{D}$ and $B$ are diagonal entries so they commute, while the third equation holds because $\tilde{D}^2 = I_n$.  $\square$

## A.9  Proof of Theorem 2.5

We can take

$$
S = \begin{pmatrix} s_1 & 0 & 0 \\ 0 & \ddots & 0 \\ 0 & 0 & s_n \end{pmatrix},
$$

which is an $n \times n$ diagonal matrix and $s_i$-s are iid random variables with $\mathbb{P}\left[s_i = 1\right] = \frac{r}{n}$ and $\mathbb{P}\left[s_i = 0\right] = 1 - \frac{r}{n}$. Since $s_i^2 = s_i$, we have $S^2 = S$, hence

$$
\begin{aligned}
VE(\hat{\beta}_s, \hat{\beta}) &= \frac{\mathbb{E}\left[\operatorname{tr}[(X^\top SX)^{-1}]\right]}{\operatorname{tr}[(X^\top X)^{-1}]} \\
PE(\hat{\beta}_s, \hat{\beta}) &= \frac{\mathbb{E}\left[\operatorname{tr}[(X^\top SX)^{-1}X^\top X]\right]}{p}.
\end{aligned}
$$

Since $X$ is unitarily invariant and $S$ is a diagonal matrix independent from $X$, $\{S, X, X^\top\}$ are almost surely asymptotically free in the non-commutative probability space by Theorem 4.3.11 of Hiai and Petz [2006]. Since the law of $S$ converges to $\mu_S = \xi\delta_1 + (1 - \xi)\delta_0$, the law of $X$ converges to $\mu_X$, thus the law of $SXX^\top S$ converges to the freely multiplicative convolution $\mu_S \boxtimes \mu_X$. The rest of the proof is the same as that in the proof of Theorem 2.4.

## A.10  Proof of Theorem 2.6

Define

$$
S = \begin{pmatrix} s_1 & & \\ & \ddots & \\ & & s_n \end{pmatrix}, \quad W = \begin{pmatrix} w_1 & & \\ & \ddots & \\ & & w_n \end{pmatrix},
$$

where the $s_i$-s are independent and $s_i|\pi_i \sim Bernoulli(\pi_i)$. $S$ is independent of $Z$ because $\pi_i$ is independent of $z_i$, by the assumption. $W$ has l.s.d. $F_w$. According to Proposition A.1, the values of

$VE$, $PE$ are determined by $\mathrm{tr}[(X^\top X)^{-1}]$, $\mathrm{tr}[Q_1(S,X)] = \mathrm{tr}[(X^\top SX)^{-1}]$, and $\mathrm{tr}[Q_2(S,X)]$. Note that under the elliptical model $X = WZ\Sigma^{1/2}$, we have

$$\mathrm{tr}[(X^\top X)^{-1}] = \mathrm{tr}[(\Sigma^{1/2}Z^\top W^2 Z\Sigma^{1/2})^{-1}],$$
$$\mathrm{tr}[Q_1(S,X)] = \mathrm{tr}[(\Sigma^{1/2}Z^\top WSWZ\Sigma^{1/2})^{-1}],$$
$$\mathrm{tr}[Q_2(S,X)] = \mathrm{tr}[(Z^\top WSWZ)^{-1}Z^\top W^2 Z].$$

Note that the e.s.d. of $\Sigma$ converges in distribution to some probability distribution $F_\Sigma$, and the e.s.d. of $WSW$ converges in distribution to $F_{sw^2}$, the limiting distribution of $s_i w_i^2$, $i = 1, \ldots, n$. Again from the results of Zhang [2007] or Paul and Silverstein [2009], with probability 1, the e.s.d. of $C_n = \frac{1}{n}\Sigma^{1/2}Z^\top WSWZ\Sigma^{1/2}$ converges to a probability distribution function $F_C$, whose Stieltjes transform $m_C(z)$, for $z \in \mathbb{C}^+$ is given by

$$m_C(z) = \int \frac{1}{t\int \frac{u}{1+\gamma e_C u}dF_{sw^2}(u) - z}dF_\Sigma(t),$$

where $e_C = e_C(z)$ is the unique solution in $\mathbb{C}^+$ of the equation

$$e_C = \int \frac{t}{t\int \frac{u}{1+\gamma e_C u}dF_{sw^2}(u) - z}dF_\Sigma(t).$$

Similarly, the e.s.d. of $D_n = \frac{1}{n}\Sigma^{1/2}Z^\top W^2 Z\Sigma^{1/2}$ converges to a probability distribution $F_D$, whose Stieltjes transform $m_D(s)$, for $z \in \mathbb{C}^+$ is given by

$$m_D(z) = \int \frac{1}{t\int \frac{u}{1+\gamma e_D u}dF_{w^2}(u) - z}dF_\Sigma(t),$$

where $e_D = e_D(z)$ is the unique solution in $\mathbb{C}^+$ of the equation

$$e_D = \int \frac{t}{t\int \frac{u}{1+\gamma e_D u}dF_{w^2}(u) - z}dF_\Sigma(t).$$

Since $F_C$ and $F_D$ have no point mass at the origin, we can set $z = 0$ Couillet and Hachem [2014]. Therefore

$$m_C(0) = \frac{1}{\int \frac{u}{1+\gamma e_C(0)u}dF_{sw^2}(u)}\int \frac{1}{t}dF_\Sigma(t), \quad e_C(0) = \frac{1}{\int \frac{u}{1+\gamma e_C(0)u}dF_{sw^2}(u)}.$$

Note also that

$$e_C(0) = \frac{\gamma e_C(0)}{\int \frac{\gamma e_C(0)u}{1+\gamma e_C(0)u}dF_{sw^2}(u)} = \frac{\gamma e_C(0)}{1 - \eta_{sw^2}(\gamma e_C(0))},$$

thus $\eta_{sw^2}(\gamma e_C(0)) = 1 - \gamma$, and

$$m_C(0) = e_C(0)\int \frac{1}{t}dF_\Sigma(t) = \frac{\eta_{sw^2}^{-1}(1-\gamma)}{\gamma}\int \frac{1}{t}dF_\Sigma(t). \tag{A.21}$$

Similarly,

$$m_D(0) = e_D(0) \int \frac{1}{t} dF_\Sigma(t) = \frac{\eta_{w^2}^{-1}(1-\gamma)}{\gamma} \int \frac{1}{t} dF_\Sigma(t).$$

Hence, again by the same argument as we have seen several times before, the traces have limits that can be evaluated in terms of Stieltjes transforms, and we have

$$VE(\hat{\beta}_s, \hat{\beta}) = \frac{\mathrm{tr}[Q_1(S,X)]}{\mathrm{tr}[(X^\top X)^{-1}]} = \frac{\mathrm{tr}[(\Sigma^{1/2} Z^\top W S W Z \Sigma^{1/2})^{-1}]}{\mathrm{tr}[(\Sigma^{1/2} Z^\top W^2 Z \Sigma^{1/2})^{-1}]}$$

$$\to \frac{m_C(0)}{m_D(0)} = \frac{\eta_{sw^2}^{-1}(1-\gamma)}{\eta_{w^2}^{-1}(1-\gamma)},$$

and the result for $VE$ follows.

We then deal with $PE$. Note that

$$PE(\hat{\beta}_s, \hat{\beta}) = \frac{\mathbb{E}\left[\mathrm{tr}[(Z^\top W S W Z)^{-1} Z^\top W^2 Z]\right]}{p}.$$

We first assume that $Z$ has iid $\mathcal{N}(0,1)$ entries. Denote $T_1 = WSW$, $T_2 = W(I-S)W$. Since $S$ is a diagonal matrix whose diagonal entries are 1 or 0, W is also a diagonal matrix, $T_1$ and $T_2$ are both diagonal matrices and the set of their nonzero entries is complementary. So $Z^\top T_1 Z$ and $Z^\top T_2 Z$ are independent from each other and $T_1 + T_2 = W^2$. We have

$$\mathbb{E}\left[\mathrm{tr}[(Z^\top W S W Z)^{-1} Z^\top W^2 Z]\right] = \mathbb{E}\left[\mathrm{tr}[(Z^\top T_1 Z)^{-1} Z^\top (T_1 + T_2) Z]\right]$$

$$= \mathbb{E}\left[\mathrm{tr}[I_p + (Z^\top T_1 Z)^{-1} Z^\top T_2 Z]\right]$$

$$= p + \mathrm{tr}[\mathbb{E}\left[(Z^\top T_1 Z)^{-1}\right] \mathbb{E}\left[Z^\top T_2 Z\right]].$$

Note that

$$\mathbb{E}\left[(Z^\top T_2 Z)_{ij}\right] = \sum_{k=1}^n \mathbb{E}\left[z_{ki} T_{2,kk} z_{kj}\right] = \sum_{k=1}^n T_{2,kk} \delta_{ij}$$

thus

$$\mathbb{E}\left[Z^\top T_2 Z\right] = \mathbb{E}\left[\mathrm{tr}(T_2)\right] I_p,$$

$$\mathbb{E}\left[\mathrm{tr}[(Z^\top W S W Z)^{-1} Z^\top W^2 Z]\right] = p + \mathbb{E}\left[\mathrm{tr}(T_2)\right] \mathrm{tr}[\mathbb{E}\left[(Z^\top T_1 Z)^{-1}\right]].$$

Note that $\frac{1}{n} Z^\top W S W Z$ is equal to $C_n$ with $\Sigma$ replaced by the identity. Thus by (A.21),

$$\frac{1}{p} \mathrm{tr}[(\frac{1}{n} Z^\top W S W Z)^{-1}]] \xrightarrow{a.s.} \frac{\eta_{sw^2}^{-1}(1-\gamma)}{\gamma},$$

$$\mathrm{tr}[(Z^\top W S W Z)^{-1}]] \xrightarrow{a.s.} \eta_{sw^2}^{-1}(1-\gamma),$$

thus

$$\lim_{n\to\infty} PE(\hat{\beta}_s, \hat{\beta}) = 1 + \frac{1}{p} \mathrm{tr}(T_2) \eta_{sw^2}^{-1}(1-\gamma)$$

$$= 1 + \frac{1}{\gamma} \mathbb{E}\left[w^2(1-s)\right] \eta_{sw^2}^{-1}(1-\gamma)$$

Then we use a similar Lindeberg swapping argument as in Theorem 2.3 to show extend this to $Z$ with iid entries of zero mean, unit variance and finite fourth moment. This finishes the proof for $PE$. For the last claim, for $OE$, note that

$$
\begin{aligned}
\mathbb{E}\left[x_t^\top (X^\top X) x_t\right] &= \mathbb{E}\left[w^2\right] \mathbb{E}\left[z_t^\top (Z^\top W^2 Z)^{-1} z_t\right] \\
&= \mathbb{E}\left[w^2\right] \mathbb{E}\left[\mathrm{tr}[(Z^\top W^2 Z)^{-1}]\right] \\
&\to \mathbb{E}\left[w^2\right] \eta_{w^2}^{-1}(1-\gamma),
\end{aligned}
$$

and that

$$
\begin{aligned}
\mathbb{E}\left[x_t^\top (X^\top S^\top S X) x_t\right] &= \mathbb{E}\left[w^2\right] \mathbb{E}\left[z_t^\top (Z^\top W S W Z)^{-1} z_t\right] \\
&= \mathbb{E}\left[w^2\right] \mathbb{E}\left[\mathrm{tr}[(Z^\top W S W Z)^{-1}]\right] \\
&\to \mathbb{E}\left[w^2\right] \eta_{sw^2}^{-1}(1-\gamma).
\end{aligned}
$$

Thus

$$
\lim_{n\to\infty} OE(\hat{\beta}_s, \hat{\beta}) = \lim_{n\to\infty} \frac{1 + \mathbb{E}\left[x_t^\top (X^\top S^\top S X)^{-1} x_t\right]}{1 + \mathbb{E}\left[x_t^\top (X^\top X)^{-1} x_t\right]} = \frac{1 + \mathbb{E}\left[w^2\right] \eta_{sw^2}^{-1}(1-\gamma)}{1 + \mathbb{E}\left[w^2\right] \eta_{w^2}^{-1}(1-\gamma)},
$$

This finishes the proof.

**Proof of leverage sampling**

It suffices to show that leverage score sampling that samples the $i$-th row with probability $\min(\frac{r}{p} h_{ii}, 1)$ is equivalent to sample with probability $\min\left[\frac{r}{p}\left(1 - \frac{1}{1 + w^2 \eta_{w^2}^{-1}(1-\gamma)}\right), 1\right]$. Given that the latter probability is independent from $z_i$, the statement of the corollary will then follow directly from Theorem 2.6.

To see this equivalence, first note that

$$
\begin{aligned}
h_{ii} &= x_i^\top \Big(\sum_{j\neq i} x_j x_j^\top + x_i x_i^\top\Big)^{-1} x_i = x_i^\top \Big(\sum_{j\neq i} x_j x_j^\top\Big)^{-1} x_i - \frac{(x_i^\top (\sum_{j\neq i} x_j x_j^\top)^{-1} x_i)^2}{1 + x_i^\top (\sum_{j\neq i} x_j x_j^\top)^{-1} x_i} \\
&= \frac{x_i^\top (\sum_{j\neq i} x_j x_j^\top)^{-1} x_i}{1 + x_i^\top (\sum_{j\neq i} x_j x_j^\top)^{-1} x_i}
\end{aligned}
$$

and

$$
\begin{aligned}
\frac{1}{1 - h_{ii}} &= 1 + x_i^\top \Big(\sum_{j\neq i} x_j x_j^\top\Big)^{-1} x_i = 1 + w_i^2 z_i^\top \Sigma^{1/2} \Big(\sum_{j\neq i} x_j x_j^\top\Big)^{-1} \Sigma^{1/2} z_i \\
&= 1 + w_i^2 z_i^\top \Big(\sum_{j\neq i} w_j^2 z_j z_j^\top\Big)^{-1} z_i.
\end{aligned}
$$

Denote $R = \sum_{j=1}^n w_j^2 z_j z_j^\top$, $R_{(i)} = \sum_{j\neq i} w_j^2 z_j z_j^\top$, so that $\frac{1}{1 - h_{ii}} = 1 + w_i^2 z_i^\top R_{(i)}^{-1} z_i$.

Since $z_i$ and $R_{(i)}$ are independent for each $i = 1, \ldots, n$, while $z_i$ has iid entries of zero mean and unit variance and bounded moments of sufficienctly high order, then by the concentration of quadratic forms lemma A.11 cited below, we have

$$
\frac{1}{n} z_i^\top R_{(i)}^{-1} z_i - \frac{1}{n} \mathrm{tr}(R_{(i)}^{-1}) \xrightarrow{a.s.} 0.
$$

**Lemma A.11** (Concentration of quadratic forms, consequence of Lemma B.26 in Bai and Silverstein [2010])**.** *Let $x \in \mathbb{R}^p$ be a random vector with iid entries and $\mathbb{E}\left[x\right] = 0$, for which $\mathbb{E}\left[(\sqrt{p}x_i)^2\right] = \sigma^2$ and $\sup_i \mathbb{E}\left[(\sqrt{p}x_i)^{4+\eta}\right] < C$ for some $\eta > 0$ and $C < \infty$. Moreover, let $A_p$ be a sequence of random $p \times p$ symmetric matrices independent of $x$, with uniformly bounded eigenvalues. Then the quadratic forms $x^\top A_p x$ concentrate around their means at the following rate*

$$P(|x^\top A_p x - p^{-1}\sigma^2 \operatorname{tr} A_p|^{2+\eta/2} > C) \leq Cp^{-(1+\eta/4)}.$$

To use lemma A.11, we only need to guarantee that the smallest eigenvalue of $R_{(i)}$ is uniformly bounded below. For this, it is enough that the smallest eigenvalue of $R$ is uniformly bounded below. Since $w_i$ are bounded away from zero, this property follows from the corresponding one for the sample covariance matrix of $z_i$, which is just the well-known Bai-Yin law [Bai and Silverstein, 2010].

Continuing with our argument, by the standard rank-one-perturbation argument [Bai and Silverstein, 2010], we have $\lim_{n\to\infty} \frac{1}{n}\operatorname{tr}[R_{(i)}^{-1}] - \frac{1}{n}\operatorname{tr}[R^{-1}] = 0$, since $R_{(i)}$ is a rank-one perturbation of $R$. Recall that $Z$ has iid entries satisfying $\mathbb{E}\left[Z_{ij}\right] = 0, \mathbb{E}\left[Z_{ij}^2\right] = 1$. Moreover, it is easy to see that by the $4 + \eta$-th moment assumption we have for each $\delta > 0$ that

$$\frac{1}{\delta^2 np}\sum_{i,j}\mathbb{E}\left[Z_{ij}^2 I_{[|Z_{ij}|>\delta\sqrt{n}]}\right] \to 0, \text{ as } n \to \infty.$$

Also, the e.s.d. of $W^2$ converges weakly to the distribution of $w^2$. By the results of Zhang [2007] or Paul and Silverstein [2009], with probability 1, the e.s.d. of $B_n = n^{-1}Z^\top W^2 Z$ converges in distribution to a probability distribution $F_B$ whose Stieltjes transform satisfies

$$m_B(z) = \frac{1}{\int \frac{s}{1+\gamma e_B s}dF_{w^2}(s) - z},$$

where for $z \in \mathbb{C}^+$, $e_B = e_B(z)$ is the unique solution in $\mathbb{C}^+$ to the equation

$$e_B = \frac{1}{\int \frac{s}{1+\gamma e_B s}dF_{w^2}(s) - z}.$$

Also, by the same reasoning as in the proof of the Haar matrix case, the l.s.d. is supported on an interval bounded away from zero. This means that we can find the almost sure limits of the traces in terms of the Stieltjes transform of the l.s.d. at zero, or equivalently in terms of the inverse eta-transform:

$$\frac{1}{p}\operatorname{tr}(\frac{1}{n}R^{-1}) = \frac{n}{p}\operatorname{tr}[(Z_w^\top Z_w)^{-1}] \to \frac{\eta_{w^2}^{-1}(1-\gamma)}{\gamma},$$

and therefore $\operatorname{tr}(R^{-1}) \xrightarrow{a.s.} \eta_{w^2}^{-1}(1-\gamma)$. Thus, from the expression of $h_{ii}$ given at the beginning, we also have

$$|h_{ii} - 1 + \frac{1}{1 + w_i^2\eta_{w^2}^{-1}(1-\gamma)}| \xrightarrow{a.s.} 0.$$

Thus as $n$ goes to infinity, leverage-based sampling is equivalent to sampling $x_i$ with probability

$$\pi_i = \min\left(\frac{r}{p}(1 - \frac{1}{1 + w_i^2 \eta_{w^2}^{-1}(1 - \gamma)}), 1\right),\tag{A.22}$$

in the sense that $\left|\min(\frac{r}{p}h_{ii}, 1) - \pi_i\right| \xrightarrow{a.s.} 0$. Therefore, it is not hard to see that the performance metrics we study have the same limits for leverage sampling and for sampling with respect to $\pi_i$. We argue for this in more detail below. Let $S^*$ be the sampling matrix based on the leverage scores, with diagonal entries $s_i^* \sim Bernoulli(\min(r/nh_{ii}, 1))$. This is the original sampling mechanism to which the theorem refers. Now, we have shown that $\|S - S^*\|_{op} \to 0$ almost surely. Because of this, one can check that $\mathrm{tr}[Q_1(S, X)] - \mathrm{tr}[Q_1(S^*, X)] \to 0$ almost surely. This follows by a simple matrix calculation expressing $A^{-1} - B^{-1} = -A^{-1}(B - A)B^{-1}$, and bounding the trace using Lemma A.7.

## A.11 Greedy leverage sampling

As a direct corollary of Theorem 2.6, we have the results for greedy leverage sampling.

**Corollary A.12** (Greedy leverage sampling). *Under the conditions of Theorem 2.6, suppose that for $p < r < n$, we take the $r$ rows of $X$ with the highest leverage scores and do linear regression on the resulting subsample of $X, Y$. Let $\tilde{w}^2 = w^2 1_{[w^2 > F_{w^2}^{-1}(1-\xi)]}$ denote the distribution of $F_{w^2}$ truncated at $1 - \xi$. Then*

$$\lim_{n\to\infty} VE(\hat{\beta}_s, \hat{\beta}) = \frac{\eta_{\tilde{w}^2}^{-1}(1 - \gamma)}{\eta_{w^2}^{-1}(1 - \gamma)},$$

$$\lim_{n\to\infty} VE(\hat{\beta}_s, \hat{\beta}) = 1 + \frac{1}{\gamma}\mathbb{E}\left[w^2 1_{[w^2 < F_{w^2}^{-1}(1-\xi)]}\right]\eta_{\tilde{w}^2}^{-1}(1 - \gamma/\xi),$$

$$\lim_{n\to\infty} OE(\hat{\beta}_s, \hat{\beta}) = \frac{1 + \mathbb{E}\left[w^2\right]\eta_{\tilde{w}^2}^{-1}(1 - \gamma)}{1 + \mathbb{E}\left[w^2\right]\eta_{w^2}^{-1}(1 - \gamma)},$$

*where $\eta_{w^2}$ and $\eta_{\tilde{w}^2}$ are the $\eta$-transforms of $F_{w^2}$ and $F_{\tilde{w}^2}$, respectively, and the expectations are taken with respect to those limiting distributions.*

## A.12 Table of tradeoff between computation and statistical accuracy

We give a summary of the algorithmic complexity and statistical accuracy (variance efficiency) of each method in Table 1.

## A.13 Simulation for leverage-based sampling

We consider a simple example where $w$ follows a discrete distribution, with $\mathbb{P}[w_i = \pm d_1] = \mathbb{P}[w_i = \pm d_2]$ $= 1/4$. $Z$ is a standard Gaussian random matrix and $\Sigma$ is the identity matrix. We plot simulation results as well as our theory for leverage score sampling, greedy leverage scores, uniform sampling and Hadamard projection. In the right panel, we also plot the histogram of the leverage scores of $X$. Our theory agrees very well with the simulations.

We also observe that the greedy leverage sampling outperforms random leverage sampling, especially for relatively small $r$. Moreover, leverage sampling and greedy leverage scores have

Table 1: Tradeoff between computation and statistical accuracy.

| Data matrix $X$ | Sketching matrix $S$ | VE | Computational complexity | Parallelization of sketching across $n$ |
|---|---|---|---|---|
| Incoherent, near-iid | Uniform sampling | $\dfrac{n-p}{r-p}$ | $O(rp^2)$ | Embarassingly parallel |
| Arbitrary | Hadamard | | $O(np\log n + rp^2)$ | Nontrivial |
| Arbitrary | iid entries | $1 + \dfrac{n-p}{r-p}$ | $O(rnp + rp^2)$ | Embarassingly parallel |

much better performances than uniform sampling. This is because the leverage scores are highly nonuniform in this example.

In Figure 2, we also compare the theoretical performance of leverage score sampling and Hadamard projection in the same elliptical model, with several aspect ratios $\gamma$ and $d_1, d_2$. We skip the comparison with Gaussian/iid projection because the performance of Hadamard projection is uniformly better, as has been shown before. The difference between $d_1$ and $d_2$ is a measure of the non-uniformity of the data.

When the data is relatively uniform (left panel), leverage sampling and Hadamard projection have similar VE. When in addition $r$ is small, leverage score sampling tends to perform better than Hadamard projection. However, when the dataset is nonuniform (right panel), leverage sampling and Hadamard projection can have very different performance. When $\gamma$ is small, leverage sampling works much better; but when $\gamma$ is large, Hadamard is uniformly better. Thus, when the dataset is nonuniform and the targeted dimension is rather small, leverage score sampling is the recommended method, provided that one can estimate the leverage scores efficiently. In conclusion, this example shows that the relative performance of sketching methods on elliptical data is quite complex, and perhaps one should mostly expect rules of thumb, instead of definitive answers.

Following is the details of the calculation.

$$\eta_{w^2}(z) = \frac{1}{2}\frac{1}{1+zd_1^2} + \frac{1}{2}\frac{1}{1+zd_2^2},$$

$$\eta_{sw^2}(z) = (1 - \frac{1}{2}\min(\pi_1,1) - \frac{1}{2}\min(\pi_2,1)) + \frac{1}{2}\min(\pi_1,1)\frac{1}{1+d_1^2 z} + \frac{1}{2}\min(\pi_2,1)\frac{1}{1+d_2^2 z},$$

where

$$\pi_1 = \frac{\xi}{\gamma}(1 - \frac{1}{1+d_1^2\eta_{w^2}^{-1}(1-\gamma)}), \quad \pi_2 = \frac{\xi}{\gamma}(1 - \frac{1}{1+d_2^2\eta_{w^2}^{-1}(1-\gamma)}).$$

It is easy to see that

$$\pi_1 + \pi_2 = 2\xi,$$

and

$$\eta_{F_{w^2}}^{-1}(1-\gamma) = \frac{1}{2d_1^2 d_2^2}(-d_1^2 - d_2^2 + \frac{d_1^2+d_2^2}{2(1-\gamma)} + \sqrt{(d_1^2+d_2^2 - \frac{d_1^2+d_2^2}{2(1-\gamma)}) + \frac{4d_1^2 d_2^2\gamma}{1-\gamma}}),$$

Figure 1: Leverage sampling, greedy leverage sampling and uniform sampling for elliptical model. We generate the data matrix $X$ from the elliptical model defined in (1), and we take $d_1 = 1, d_2 = 3, n = 20000, p = 1000$ while $Z$ is generated with iid $\mathcal{N}(0,1)$ entries and $\Sigma$ is the identity. We let $r$ range from 4000 to 20000. At each dimension $r$ we repeat the experiments 50 times and take the average. For leverage sampling, we sample each row of $X$ independently with probability $\min(r/p \cdot h_{ii}, 1)$. For greedy leverage scores, we take the $r$ rows of $X$ with the largest leverage scores. For uniform sampling, we uniformly sample $r$ rows of $X$. We see a good match between theory and simulations.

If we use the $r$ rows of $X$ with the largest leverage scores, the truncated distribution $\tilde{w}$ in Theorem A.12 can be written as

$$F_{\tilde{w}^2}(t) = \begin{cases} \delta_{d_2^2}, & 0 < \frac{r}{n} \leq \frac{1}{2} \\ (1 - \frac{n}{2r})\delta_{d_1^2} + \frac{n}{2r}\delta_{d_2^2}, & \frac{1}{2} < \frac{r}{n} \leq 1. \end{cases}$$

Therefore

$$\eta_{\tilde{w}^2}(z) = \begin{cases} \frac{1}{1+d_2^2 z}, & 0 < \frac{r}{n} \leq \frac{1}{2} \\ (1 - \frac{n}{2r})\frac{1}{1+d_1^2 z} + \frac{n}{2r}\frac{1}{1+d_2^2 z}, & \frac{1}{2} < \frac{r}{n} \leq 1, \end{cases}$$

thus

$$\eta_{F_{\tilde{w}^2}}^{-1}\left(1 - \frac{\gamma}{\xi}\right) = \begin{cases} \frac{\gamma}{d_2^2(\xi-\gamma)}, & 0 < \frac{r}{n} \leq \frac{1}{2} \\ \frac{1}{2d_1^2 d_2^2}[-b + \sqrt{b^2 + \frac{4d_1^2 d_2^2 \gamma}{\xi-\gamma}}], & \frac{1}{2} < \frac{r}{n} \leq 1. \end{cases}$$

Figure 2: Comparing leverage sampling and Hadamard projection.

Here

$$b = d_1^2 + d_2^2 - \frac{(2\xi - 1)d_2^2 + d_1^2}{2(\xi - \gamma)}.$$

## A.14 Simulation for nonuniform data

In Figure 3, each row of $X$ is generated from a $t$ distribution with 1 degree of freedom. Specifically, let $\Sigma$ be $p \times p$ covariance matrix with $\Sigma_{ij} = 2 \times 2^{-|i-j|}$. Then each row of $X$ is generated as $\mathcal{N}(0, \Sigma)$ divided by a chi-squared random variable with 1 degree of freedom. We show the mean, as well as the 5% and 95% quantiles of VE over 1000 repetitions. We do not use standard deviation to illustrate the variability, because the variance can be rather large.

We also plot the histogram of the leverage scores on the right. There are several extremely large leverage scores, which means that the design matrix is ill-conditioned. For readability's sake, we do not plot the results for uniform sampling and leverage sampling. Instead, we show them in Tables 2 and 3. We observe the following:

- Usually, the numerical mean of VE falls on the respective theoretical line. Moreover, the 95% confidence intervals always cover the theoretical lines. This means that our results are correct on average.

- However, the VE can be anomalously large in some rare cases, driving the mean to be rather large. But even among the 1000 repetitions, the anomalous values only fewer than ten times. This explains why the standard deviations are large but the 90% confidence intervals are relatively short.

Figure 3: $t$ distribution.

- The reason for the abnormal phenomena is due to some rows of $X$ with large norms, which dominate the influence of $X$ on $Y$. When sketching the matrix, we shrink the influence of these dominating rows, either by mixing with other unimportant rows or by dropping them altogether. Therefore the sketched estimators lose too much accuracy.

- Even in this less favorable situation, the Hadamard transform is still the most desirable sketching method. It has small average VE, relatively small variability (i.e., short confidence intervals), and short running time.

## A.15   OE for two empirical datasets

See Figure 4 for the out-of-sample error on the two empirical datasets: Million Song Dataset (MSD) and the Flight Dataset.

## A.16   Comparison with previous bounds

We also compare our results with the upper bounds given in Raskutti and Mahoney [2016]. For sub-Gaussian projections, they showed that if $r \geq c \log n$, then with probability greater than 0.7,

Table 2: Uniform sampling, $\log VE$

|  | mean | 5% | 95% | 50% |
|---|---|---|---|---|
| 0 | 20.25823968 | 3.09919506 | 10.90630978 | 9.689192869 |
| 1 | 17.26805584 | 2.044572843 | 9.33262643 | 7.881064556 |
| 2 | 10.78396456 | 1.768101235 | 7.978181897 | 6.650869548 |
| 3 | 11.63550561 | 3.043487316 | 7.369292418 | 5.715237875 |
| 4 | 9.203440888 | 0.794976879 | 6.595984621 | 5.029794939 |
| 5 | 8.980845283 | 1.326756793 | 5.774310548 | 4.38426635 |
| 6 | 7.380001677 | 1.381715379 | 5.809281226 | 3.871467657 |
| 7 | 5.175269082 | -0.182261694 | 5.605915977 | 3.399170722 |
| 8 | 6.359148538 | 0.113763057 | 4.175648541 | 2.948032272 |
| 9 | 9.15176239 | 1.078751974 | 4.24045247 | 2.574611614 |
| 10 | 3.947147126 | 0.814135635 | 3.999019444 | 2.265773149 |
| 11 | 3.44225402 | 0.122953471 | 3.484524911 | 1.932062314 |
| 12 | 2.527325826 | 0.800658751 | 2.987471342 | 1.6527189 |
| 13 | 1.824634546 | 0.322903628 | 2.587082545 | 1.331773852 |
| 14 | 1.491822848 | 0.325396696 | 2.165972084 | 1.07823758 |
| 15 | 1.04419024 | 0.130949401 | 1.589996964 | 0.81014184 |
| 16 | 0.934085598 | 0.088131217 | 1.387329021 | 0.596745268 |
| 17 | 0.873952782 | -0.052823305 | 1.034961439 | 0.375621926 |
| 18 | 0.447688303 | -0.130411888 | 0.61016263 | 0.17981115 |
| 19 | 6.978320808 | -0.189431268 | 0.226659269 | -4.75E-08 |

Figure 4: OE for MSD and flight dataset.

Table 3: Leverage sampling, $\log VE$

|     | mean        | 5%           | 95%         | 50%          |
|-----|-------------|--------------|-------------|--------------|
| 0   | 7.626778204 | -0.159794491 | 0.821111174 | 0.204571271  |
| 1   | 3.216688922 | -0.223342993 | 0.456512164 | 0.059604721  |
| 2   | 1.015004939 | -0.207986859 | 0.424644872 | 0.029099432  |
| 3   | 6.287155109 | -0.161735746 | 0.333680227 | 0.012818312  |
| 4   | 0.882013996 | -0.238995897 | 0.397351467 | 0.010611862  |
| 5   | 0.133943751 | -0.195882891 | 0.322395411 | 0.002172268  |
| 6   | 0.487048617 | -0.160963212 | 0.256059749 | 0.002658917  |
| 7   | 0.204928838 | -0.205892837 | 0.302131442 | 0.001159857  |
| 8   | 1.354935903 | -0.212644915 | 0.376751395 | 0.001978275  |
| 9   | 0.691138831 | -0.174171007 | 0.290014414 | 0.001378926  |
| 10  | 3.129679099 | -0.250449933 | 0.432175622 | 6.24E-05     |
| 11  | 0.40467726  | -0.280542607 | 0.403070117 | 0.000882543  |
| 12  | 3.800307066 | -0.206858102 | 0.452128865 | -0.000171639 |
| 13  | 0.403587432 | -0.18407553  | 0.251072808 | -7.39E-05    |
| 14  | 0.758228813 | -0.296238449 | 0.452796686 | 0.000292609  |
| 15  | 0.180781152 | -0.208918435 | 0.395642624 | 5.19E-05     |
| 16  | 0.075991698 | -0.115281186 | 0.202626369 | 8.07E-05     |
| 17  | 0.159661829 | -0.191824839 | 0.243641727 | 6.21E-05     |
| 18  | 1.00887942  | -0.218371065 | 0.369474928 | 3.30E-05     |
| 19  | 1.994998633 | -0.145457064 | 0.347448221 | 7.52E-08     |

it holds that

$$PE \leq 44(1 + \frac{n}{r}), \ RE \leq 1 + 44\frac{p}{r}.$$

For Hadamard projection, they showed that if $r \geq cp \log n (\log p + \log \log n)$, then with probability greater than 0.8, it holds that

$$PE \leq 1 + 40 \log(np)(1 + \frac{p}{r}), \ RE \leq 40 \log(np)(1 + \frac{n}{r}).$$

In Figure 5, we plot both our theoretical lines and the above upper bounds, as well as the simulation results. It is shown that our theory is much more accurate than these upper bounds.

## A.17 Computation time

In this section we perform a more rigorous empirical comparison of the running time of sketching. We know that the running time of OLS has order of magnitude $O(np^2)$, while the running time of Hadamard projections is $c_1 np^2 + c_2 np \log(n)$ for some constants $c_i$. While the cubic term clearly dominates for large $n, p$, our goal is to understand the performance for finite samples $n, p$ on typical commodity hardware. For this reason, we perform careful timing experiments to determine the approximate values of the constants on a MacBook Pro (2.5 GHz CPU, Intel Core i7).

We obtain the following results. The time for full OLS and Hadamard sketching is approximately

$$t_{full} = 4 \times 10^{-11} np^2, \ t_{Hadamard} = 2 \times 10^{-8} pn \log n + 4 \times 10^{-11} rp^2$$

Figure 5: Comparison with prior bounds. In this simulation, we let $n = 2000$, the aspect ratio $\gamma = 0.05$, with $r/n$ ranging from 0.15 to 1. The first column displays the results for $PE$ and $RE$ for Gaussian projection, while the second column shows results for randomized Hadamard projection. The $y$-axis is on the log scale. The data matrix $X$ is generated from Gaussian distribution and fixed at the beginning, while the coefficient $\beta$ is generated from uniform distribution and also fixed. At each dimension $r$ we repeat the simulation 50 times and all the relative efficiencies are averaged over 50 simulations. In each simulation, we generate the noise $\varepsilon$ as well as the sketching matrix $S$. The orange dotted lines are drawn according to Section A.16, while the blue dashed lines are drawn according to our Theorem 2.1 and Theorem 2.4.

See Figure 6 for a comparison of the running times for various combinations of $n, p$. For instance, we show the results for $n = 7 \cdot 10^4$, and $p = 1.4 \cdot 10^4$ with the sampling ratio ranging from 0.2 to 1. We see that we save time if we take $r/n \le 0.6$.

We can also perform a more quantitative analysis. If we want to reduce the time by a factor of $0 < c < 1$, then we need

$$\frac{2 \times 10^{-8} pn \log n + 4 \times 10^{-11} rp^2}{4 \times 10^{-11} np^2} \le c$$

or also $r \le cn - 500 \frac{n \log n}{p}$, when $0 < c - 500 \frac{\log n}{p} < 1$. Then the out-of-sample prediction efficiency is lower bounded by

$$OE(\hat{\beta}_s, \hat{\beta}) = \frac{r(n - p)}{n(r - p)}$$

$$\ge \frac{n - p}{n} \left( 1 + \frac{p}{n(c - \frac{500 \log n}{p}) - p} \right) = (1 - \gamma) \left( 1 + \frac{\gamma}{c - \frac{500 \log n}{p} - \gamma} \right).$$

This shows how much we lose if we decrease the time by a factor of $c$.

Similarly, if we want to control the $VE$, say to ensure that $VE(\hat{\beta}_s, \hat{\beta}) \le 1 + \delta$, then we need

$$r \ge \frac{n - p}{1 + \delta} + p,$$

then the we must spend at least a fraction of the full OLS time given below

$$\frac{r}{n} + \frac{500 \log n}{p} \ge \frac{1 - \gamma}{1 + \delta} + \gamma + \frac{500 \log n}{p}.$$

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

Figure 6: A comparison of the running times for various combinations of $n, p$.

Romain Couillet and Walid Hachem. Analysis of the limiting spectral measure of large random matrices of the separable covariance type. *Random Matrices: Theory and Applications*, 3(04): 1450016, 2014.

Walid Hachem. An expression for $\int log(t/\sigma^2 + 1)\mu \boxtimes \tilde{\mu}(dt)$. *unpublished*, 2008.

Fumio Hiai and Dénes Petz. *The semicircle law, free random variables and entropy*. Number 77. American Mathematical Soc., 2006.

Vladimir A Marchenko and Leonid A Pastur. Distribution of eigenvalues for some sets of random matrices. *Mat. Sb.*, 114(4):507–536, 1967.

Alexandru Nica and Roland Speicher. *Lectures on the combinatorics of free probability*, volume 13. Cambridge University Press, 2006.

Debashis Paul and Alexander Aue. Random matrix theory in statistics: A review. *Journal of Statistical Planning and Inference*, 150:1–29, 2014.

Debashis Paul and Jack W Silverstein. No eigenvalues outside the support of the limiting empirical spectral distribution of a separable covariance matrix. *Journal of Multivariate Analysis*, 100(1): 37–57, 2009.

Garvesh Raskutti and Michael W Mahoney. A statistical perspective on randomized sketching for ordinary least-squares. *The Journal of Machine Learning Research*, 17(1):7508–7538, 2016.

Antonia M Tulino, Giuseppe Caire, Shlomo Shamai, and Sergio Verdú. Capacity of channels with frequency-selective and time-selective fading. *IEEE Transactions on Information Theory*, 56(3): 1187–1215, 2010.

Dan V Voiculescu, Ken J Dykema, and Alexandru Nica. *Free random variables*. Number 1. American Mathematical Soc., 1992.

Jianfeng Yao, Zhidong Bai, and Shurong Zheng. *Large Sample Covariance Matrices and High-Dimensional Data Analysis*. Cambridge University Press, New York, 2015.

Lixin Zhang. *Spectral analysis of large dimentional random matrices*. PhD thesis, 2007.