[Reviews · NeurIPS 2019]

Reviewer 1



The addressed problem (sketching) is interesting and the authors manage to obtain new (asymptotic) theoretical results for various methods. These can help analyzing their trade-offs between computational requirements and different statistical efficiency measures. The paper is well-written and the results are presented in a simple, understandable way. The proofs, located in the supplementary material, are nicely structured and seem correct. Several widespread sketching methods are studied and the presented simulation results support the theoretical claims of the paper. Drawbacks of the paper are that it only considers the basic OLS and the results are only asymptotic. These reduce the significance of the results. For example, studying some more advanced versions of LS, such as regularized LS (ridge regression) or total least-squares (TLS), would have improved the importance of the work. Also a finite-time analysis would be more convincing. Nevertheless, the asymptotic results provide nice details in understanding the behavior of various sketching methods. Post rebuttal comments: thank you for your replies, I can accept your answers and thus I have modified my recommendation accordingly.

Reviewer 2



This paper considers statistical bounds for linear regression with random projection, a method often used to speed up computation by randomly compressing a large number of data points. They consider a model with fixed design matrix X, coefficient vector b and responses y = Xb + n where n is mean 0 noise. They then consider the statistical performance of sketching versus full ordinary least squares. Looking e.g. at the ratio of E[||tilde b_s - b||_2^2]/E[||tilde b - b||_2^2] where tilde b_s and tilde b are the estimators from sketching and OLS respectively. The main idea is to identify differences between different sketching methods like subsampled Hadamard transform Gaussian random projections, and data point subsampling, which typically achieve the same asymptotic results for various theoretical guarantees up to constants but may be observed to perform differently in practice. They succeed in doing this, showing that random projections with orthogonal rows (subsampled Hadamard and Haar projections) perform better than random projections with iid entries, like Gaussian entries. The gap in performance is rather small if the sketch size is much smaller than the original input size, which seems to be the natural setting in that it would be worthwhile to apply sketching. Still though, the separation is interesting and the first theoretical separation between these methods that I am aware of. To achieve this separation, the work considerers the setting where the number of data points n, the number of parameters p (columns of X), and the sketch size r all go to infinity, with n/p and n/r converging to constants. Bounds in this asymptotic regime are also given for uniform sampling and leverage score based sampling, although the assumptions made/bounds given are not really comparable to the other methods. Generally the paper is very well written. The main body is mostly a summary of results without intuition given for why they should hold/what the proof techniques are. I think this should be reprioritized, especially since this is largely a theoretical result. The proofs for the Gaussian case are very simple and could be sketched in the main body. The proofs of the other cases use lots of seemingly interesting tools that I have not seen applied to the analysis of sketching methods before -- Stieltjes transform, free probability, eta-transform etc. I believe that introducing these tools to those working on the analysis of sketching algorithms could be a valuable contribution. However, in the current presentation, it is hard for a non expert to understand the intuition behind why certain tools are being used, how they are allowing for the results to be proven etc. E.g. how intuitively is the gap between Gaussian and subsampled Hadamard able to be shown? Beyond the very superficial 'Hadamard has orthogonal rows'. Comments: - 'we get more accurate results for the performance of sketching' -- what exactly does this mean? How is accuracy measured? - Maybe this is common in statistics but why is is reasonable to assume that p and n would grow together with the aspect ratio converging? Why do we inherently expect more parameters with more data points? - Table 1 is very hard to read. Not sure if lines are divisions or line breaks. Many entries are missing and alignments makes it impossible to tell which values are supposed to fall in which column. Variables in leverage score line are undefined. - What talking about comparison to Raskutti and Mahoney in related work what sketching method does the 1 + 44p/r term apply to? Also what are the stronger assumptions? I don't see any nontrivial assumptions listed for the random projection case in the table or in Theorem 2.1. - Was a nice illustration that for Gaussian random projections, even when r = n VE goes to 2 not 1. - In Theorem 2.3 what does it mean that X's esd converges if X is just a matrix? Doe you mean that there is a function mapping n and p to deterministic matrices and the esps of these matrices converge as n and p increase? - Line 233: what does it mean for a matrix to be orthogonally invariant? - The rotationally invariant assumption seems super strong. This isn't even the case when each row is a p-dimensional Gaussian with some fixed covariance Sigma right? - Line 255: what does it mean it doesn't 'introduce enough randomness?' How is this measured? - I don't see how the line in 280 gives leverage score sampling. Where does the eta-transform come up? Would be helpful to explain a bit more. - The experiments look quite good, especially the real world data set. Although would be a lot more convincing if more than just a single dataset were tested. - The synthetic X is just iid Gaussian which usually makes everything look really good so this part of the experiment doesn't seem to add much. I guess it does confirm the theory for uniform sampling, under the rotationally invariant assumption. What about at least X with Gaussian rows with a non identity covariance? - Was leverage score sampling just not tested because you couldn't compute the theoretical bounds in closed form? - Under Prop A.1, what is eps_t? Under Proof of Theorem 2.1 tr((X^T X)^-1) shouldn't have an expectation around it. X is fixed here right?

Reviewer 3



## Originality ## This paper studies the asymptotic behavior of sketching methods for the least square problem. The difference between this work and previous works is relatively obvious. ## Quality ## The paper is technically sound. The results of the theorems make sense and each of the theorems is fully explained. However, I did not carefully check the proof of the theorems. ## Clarity ## The paper is well written, the organization is very clear, and the notations are concise and accurate.

[Author Response · NeurIPS 2019]

We thank the reviewers for their careful reading of the manuscript, and for their constructive comments. We feel that the reviews are largely positive. In the remainder, we want to address some of the issues raised, and we will address them in detail in the revision:

Reviewer 1: "Could you comment on the possibility to generalize these results, for example, to regularized LS, LASSO, elastic net regularization, TLS, etc.?" *Generalizing to ridge regression is possible using random matrix theory, but very nontrivial, and in fact we are currently pursuing this. In our view, this would be a different project. Generalizing to Lasso, elastic net, TLS seems like a much more difficult problem. It may be possible to handle lasso using approximate message passing (AMP), but this would be a new project.*

"In the simulation experiment the input matrix X should be studied for other distributions." *We have simulations with correlated t-distributed data in Appendix A.14*

"Also a finite-time analysis would be more convincing." *For Gaussian and iid projections, it may be possible to obtain convergence rates using known results on the convergence rate of Stieltjes transforms. However, for Hadamard transforms, the only results we are aware of are asymptotic, as they are based on free probability theory. Thus finite-sample results may be hard.*

Reviewer 2: "'we get more accurate results for the performance of sketching' – what exactly does this mean? How is accuracy measured? *It means that our results are more accurate in simulations, and "get the right constant". See Appendix 16.*

Maybe this is common in statistics but why is it reasonable to assume that p and n would grow together with the aspect ratio converging? Why do we inherently expect more parameters with more data points? *This is actually just a model for "large n, large p". We do not really think that the number of parameters is growing.*

Table 1 is very hard to read. Not sure if lines are divisions or line breaks. Many entries are missing and alignments makes it impossible to tell which values are supposed to fall in which column. Variables in leverage score line are undefined. *We have made the table easier to read: add separators, copy values in multiple columns.*

What talking about comparison to Raskutti and Mahoney in related work what sketching method does the 1 + 44p/r term apply to? Also what are the stronger assumptions? I don't see any nontrivial assumptions listed for the random projection case in the table or in Theorem 2.1. *It refers to their subsampling and subgaussian projection results. The stronger conditions refer to subsampling, when we need ortho-invariance*

In Theorem 2.3 what does it mean that X's esd converges if X is just a matrix? Doe you mean that there is a function mapping n and p to deterministic matrices and the esps of these matrices converge as n and p increase? *We mean the esd of $X^T X$*

Line 233: what does it mean for a matrix to be orthogonally invariant? *This is the same as rotationally invariant, defined in lines 239-242*

The rotationally invariant assumption seems super strong. This isn't even the case when each row is a p-dimensional Gaussian with some fixed covariance Sigma right? *Indeed, this condition is quite strong and does not hold for correlated Gaussians. However, it seems that the current proof technique (asymptotic freeness, Theorem 4.3.11 of Hiai and Petz (2006)) requires it*

Line 255: what does it mean it doesn't 'introduce enough randomness?' How is this measured? *This is an intuitive claim and we do not know how to measure it*

I don't see how the line in 280 gives leverage score sampling. Where does the eta-transform come up? Would be helpful to explain a bit more. *We will explain more, but the eta-transform is the limit of the leverage score*

Although would be a lot more convincing if more than just a single dataset were tested. *Figure 2 has two datasets*

What about at least X with Gaussian rows with a non identity covariance? *We have simulations with correlated t-distributed data in Appendix A.14, "Simulation for nonuniform data"*

Was leverage score sampling just not tested because you couldn't compute the theoretical bounds in closed form? *See Appendix 13 for simulations with leverage scores*

Under Prop A.1, what is $eps_t$? Under Proof of Theorem 2.1 $tr((X^T X)^{-1})$ shouldn't have an expectation around it. X is fixed here right? $eps_t$ *is test noise. Indeed no expectation*

"I think the biggest thing is to push forward the proof techniques and give some intuition. Too much of the paper is spent just describing the results and explaining why they are good. More should be spent actually explaining the results." *We will work on explaining the results and proof techniques, and giving intuition.*

[Meta-Review · NeurIPS 2019]

Solid paper with nice separation between different sketching methods. All reviewers are in agreement that it is a solid accept.